# A HARD-TO-BEAT BASELINE FOR TRAINING-FREE CLIP-BASED ADAPTATION

**Zhengbo Wang**[1,2]  **Jian Liang**[2,3*]  **Lijun Sheng**[1,2]  **Ran He**[2,3]  **Zilei Wang**[1]  **Tieniu Tan**[2,4]

[1] University of Science and Technology of China
[2] CRIPAC & MAIS, Institute of Automation, Chinese Academy of Sciences (CASIA)
[3] School of Artificial Intelligence, University of Chinese Academy of Sciences [4] Nanjing University
`zhengbowang@mail.ustc.edu.cn, liangjian92@gmail.com`

## ABSTRACT

Contrastive Language-Image Pretraining (CLIP) has gained popularity for its remarkable zero-shot capacity. Recent research has focused on developing efficient fine-tuning methods, such as prompt learning and adapter, to enhance CLIP's performance in downstream tasks. However, these methods still require additional training time and computational resources, which is undesirable for devices with limited resources. In this paper, we revisit a classical algorithm, Gaussian Discriminant Analysis (GDA), and apply it to the downstream classification of CLIP. Typically, GDA assumes that features of each class follow Gaussian distributions with identical covariance. By leveraging Bayes' formula, the classifier can be expressed in terms of the class means and covariance, which can be estimated from the data without the need for training. To integrate knowledge from both visual and textual modalities, we ensemble it with the original zero-shot classifier within CLIP. Extensive results on 17 datasets validate that our method surpasses or achieves comparable results with state-of-the-art methods on few-shot classification, imbalanced learning, and out-of-distribution generalization. In addition, we extend our method to base-to-new generalization and unsupervised learning, once again demonstrating its superiority over competing approaches. Our code is publicly available at https://github.com/mrflogs/ICLR24.

## 1 INTRODUCTION

Contrastive Language-Image Pretraining, known as CLIP (Radford et al., 2021), has attracted considerable attention in recent years as a powerful method for aligning vision and language representations. By leveraging a massive dataset of 400 million web-scale image-text pairs, CLIP learns to encode images and text into a shared semantic space using vision and language encoders, respectively. This shared semantic space facilitates the comparison of similarities and differences between images and texts. One remarkable feature of CLIP is its zero-shot ability to perform image classification without any additional training on the target classes. This is accomplished by using the language encoder to generate classifier weights based on a simple prompt, such as "a photo of a {class}". By inputting different class names into the prompt, we can generate the corresponding weights for classification, thereby enabling the classification of images into a broad range of categories without training.

Despite its powerful zero-shot capability, recent works (Zhou et al., 2022b; Zhang et al., 2022; Zhou et al., 2022a; Lu et al., 2022; Gao et al., 2024) have focused on designing efficient fine-tuning methods for downstream classification tasks. These methods have achieved significant improvements compared to Zero-Shot CLIP (Radford et al., 2021), even when trained on few-shot datasets and optimizing extremely small numbers of parameters. For instance, CoOp (Zhou et al., 2022b) proposes to learn a set of global text prompts for the pre-trained CLIP (Radford et al., 2021) on downstream tasks, which achieves a 15% improvement compared to Zero-Shot CLIP (Radford et al., 2021) with only 16 samples per class by fine-tuning a mere 16k parameters. While these methods are efficient

---

*Corresponding author.

and yield satisfactory results in downstream tasks, they demand extra computational resources to learn new parameters. This can be inconvenient, especially on devices with limited resources.

We aim to develop a method that not only eliminates the need for additional training, similar to Zero-Shot CLIP but also attains comparable or even better results than these training-required methods. To achieve this, we revisit a classical algorithm, Gaussian Discriminant Analysis (GDA) (Bishop, 2006), and apply it to the downstream classification tasks of CLIP. Typically, GDA assumes that features from different classes follow Gaussian distributions with identical covariance. By leveraging Bayes' formula, the classification probability, $p(y|x)$, can be expressed as a softmax of a linear function, where the weight and bias are determined by the mean vectors and covariance. Based on this, we can compute the mean vectors and covariance from the training dataset to construct the classifier, thereby eliminating the need for any additional training process such as SGD. To fully utilize the knowledge in pretrained CLIP, we ensemble it with the CLIP zero-shot classifier, integrating the knowledge from both visual and textual modalities.

To further demonstrate its efficacy, we develop two variants of our method for base-to-new generalization and unsupervised learning, where our method can not be directly applied. In the base-to-new generalization, the model is trained on a base dataset and then adapted to a new dataset with different classes. Based on the observation that similar samples have similar statistical information (Yang et al., 2021), we propose using the K-Nearest-Neighbor algorithm to synthesize data for new classes, and we obtain the classifier for new classes using these synthesized data. For unsupervised learning, where no labeled data are available for training, the mean and covariance cannot be directly obtained from data. Based on the Gaussian assumption in GDA, the unlabeled data then follow a Gaussian mixture distribution. We straightforwardly employ the EM algorithm to estimate its mean and covariance. In both scenarios, we endeavor to keep the modifications relatively straightforward to avoid introducing additional complexity that might impact the assessment of our approach. Nevertheless, these two simple variants still achieve comparable performance with previous complex state-of-the-art methods.

We conduct extensive experiments on 17 widely adopted datasets to evaluate the effectiveness of our method. Despite the simplicity of our approach, we demonstrate that our method serves as a hard-to-beat baseline. In few-shot classification, our method surpasses state-of-the-art training-free methods by a margin of 2.82% on average over 11 datasets, and it achieves comparable performance with training-required methods (76.05% vs. 75.83%). For imbalanced learning, our method outperforms previous state-of-the-art methods, even if they are fully fine-tuned. The two variants of base-to-new generalization and unsupervised learning achieve comparable performance with previous methods. These results underscore the effectiveness of our method.

## 2 RELATED WORK

**Vision-Language Models.** In recent years, vision-language models (VLMs) have become a new paradigm for foundational models that aim to bridge the gap between the modalities of vision and language. These models are trained on large-scale image-text datasets, which endows them with powerful transferable abilities such as zero-shot learning, few-shot adaptation, and in-context learning (Radford et al., 2021; Kim et al., 2021; Lu et al., 2019; Su et al., 2020; Jia et al., 2021). Moreover, they exhibit strong open-world capabilities and have been successfully applied to recognize open-world concepts, including zero-shot learning (Radford et al., 2021; Jia et al., 2021), open-world segmentation (Mengde et al., 2022; Ding et al., 2022), and open-world detection (Joseph et al., 2021; Gu et al., 2022; Gupta et al., 2022). Contrastive-based vision-language pre-training has become the mainstream approach in this field. These methods, including CLIP (Radford et al., 2021) and ALIGN (Jia et al., 2021), are trained on large-scale web-based noisy image-text pairs. They employ a language encoder and a vision encoder to encode the texts and images, respectively, and learn to align their representations through contrastive loss. We utilize CLIP (Radford et al., 2021) in this work.

**Efficient Fine-tuning for VLMs.** Recent works (Zhou et al., 2022a;b; Zhang et al., 2022; Gao et al., 2024; Lu et al., 2022; Chen et al., 2023; Guo et al., 2023; Udandarao et al., 2023; Huang et al., 2022; Wang et al., 2024; 2023b) focus on developing efficient fine-tuning methods for large pre-trained vision-language models that can be used in downstream tasks due to their large model size. These methods aim to achieve the maximum performance gain by fine-tuning the minimum number of model parameters on few-shot downstream datasets. For instance, CoOp (Zhou et al., 2022b) proposes to learn global text prompts for downstream tasks through back-propagation on few-shot datasets.

Meanwhile, CLIP-Adapter (Gao et al., 2024) proposes to learn a visual and a textual adapter to refine the original representations of the vision-language models. Despite being efficient and achieving significant improvements, previous works (Zhou et al., 2022a; Wang et al., 2023b) have found that these methods tend to exhibit poor generalization when confronted with new classes. In response to this limitation, CoCoOp (Zhou et al., 2022a) proposes a solution by incorporating visual information into the text prompt for regularization, leading to improved base-to-new generalization performance. Besides, other works (Wang et al., 2024; Huang et al., 2022; Manli et al., 2022) attempt to adapt CLIP (Radford et al., 2021) for imbalanced learning and unsupervised learning scenarios. While these approaches have achieved satisfactory results, they still require additional training processes to learn the newly introduced parameters, which is undesirable for devices with limited resources.

**High-dimensional Covariance Estimation.** Our method involves estimating the covariance or precision matrices in high-dimensional space, which can be challenging, particularly when data is limited. Typically, the covariance matrix is estimated using the Maximum Likelihood Estimator (MLE), but this is not a reliable estimator of the eigenvalues of the covariance matrix. As a result, the precision matrix obtained from the inversion of the estimated covariance matrix may not be accurate. In some cases, it may even be impossible to invert the empirical covariance matrix due to numerical issues. To address this problem, previous works have proposed shrinkage methods (Ledoit & Wolf, 2004; Chen et al., 2010; Kubokawa & Srivastava, 2008; Efron & Morris, 1976). For instance, Ledoit-Wolf shrinkage (Ledoit & Wolf, 2004) provides a formula to calculate the optimal shrinkage coefficient that minimizes the MSE between the estimated and actual covariance matrices. Similarly, OAS (Chen et al., 2010) presents a formula that aims to select a shrinkage coefficient that results in a lower MSE than the one given by Ledoit-Wolf shrinkage (Ledoit & Wolf, 2004). However, these approaches require additional optimization processes to estimate the covariance and precision matrices. In our work, we use the empirical Bayes ridge-type estimator (Kubokawa & Srivastava, 2008), which does not require any training, to address this challenge.

## 3 METHOD

### 3.1 GAUSSIAN DISCRIMINANT ANALYSIS FOR CLIP ADAPTATION

**Gaussian Discriminant Analysis.** We revisit a traditional probabilistic generative model (Bishop, 2006), Gaussian Discriminant Analysis (GDA), for training-free CLIP-based adaptation, whose classifier can be derived by making an assumption about the data distribution of each class. The parameters of the classifier can be obtained from the statistical information of the data without the need for explicit training. By applying Bayes' formula, the classification probability can be expressed as the function of the data distribution and its prior probability:

$$p(y=i|x) = \frac{p(x|y=i)p(y=i)}{\sum_{j=1}^{K} p(x|y=j)p(y=j)} = \frac{\exp(f_i(x))}{\sum_{j=1}^{K} \exp(f_j(x))}, \quad (1)$$

where $i = 1, 2, \ldots, K$ for $K$-class classification tasks, $x \in \mathbb{R}^D$ is the visual feature, and we normalize Eq. (1) using the softmax function. And the logit function is $f_i(x) = \log(p(x|y=i)p(y=i)), i = 1, 2, \ldots, K$. Therefore, by assuming the data distribution of each class and their prior distribution, we can obtain the classifier. In GDA (Bishop, 2006), the features are typically assumed to follow the Gaussian distributions with identical covariance, i.e., $(X|Y=i) \sim \mathcal{N}(\mu_i, \Sigma)$ for $i = 1, 2, .., K$. We substitute this assumption into Eq. (1), which then can be expressed as follows:

$$p(y=i|x) = \frac{\exp(\mu_i^T \Sigma^{-1} x - \frac{1}{2}\mu_i^T \Sigma^{-1} \mu_i + \log p_i)}{\sum_{j=1}^{K} \exp(\mu_j^T \Sigma^{-1} x - \frac{1}{2}\mu_j^T \Sigma^{-1} \mu_j + \log p_j)}, \quad (2)$$

where $p_i = p(y=i) = 1/K, i = 1, 2, ..., K$ is the prior probability of the corresponding class, which is assumed to be uniform. And thus, the logit $f_i(x) = \mu_i^T \Sigma^{-1} x - \frac{1}{2}\mu_i^T \Sigma^{-1} \mu_i + \log p_i$. Thus, the weight $W \in \mathbb{R}^{K \times D}$ and the bias $b \in \mathbb{R}^K$ for the classifier are as follows:

$$w_i = \Sigma^{-1}\mu_i, \quad b_i = \log p_i - \frac{1}{2}\mu_i^T \Sigma^{-1}\mu_i, \quad (3)$$

for $i = 1, 2, ..., K$. Later, we estimate the mean for each class and the precision matrix using the training data and subsequently obtain the corresponding weight and bias for the linear classifier.

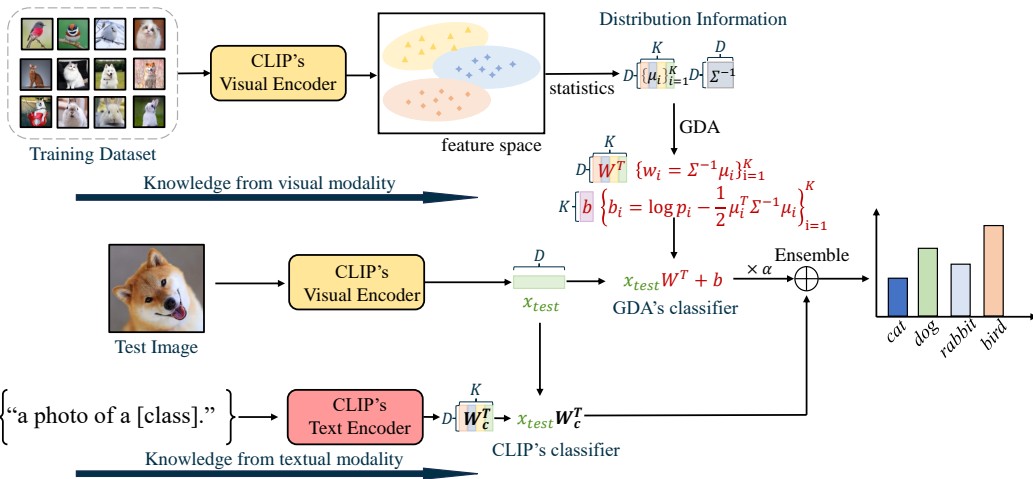

Figure 1: **The overview of our training-free method.** In our method, we begin by extracting visual features from the training dataset using the CLIP visual encoder. Next, we compute the mean vectors for each class and the shared precision matrix (inverse covariance) using Eq. (4). Through the Gaussian Discriminate Analysis (GDA), the weight and bias of the classifier can be expressed in terms of the mean vectors and the precision matrix, which can be derived from Eq. (3) (the red formula in the figure). Finally, we enhance our method by ensembling the GDA classifier and the CLIP's zero-shot classifier, integrating the knowledge from visual and textual modalities.

**Parameter Estimation.** We estimate the mean vectors using the empirical mean $\hat{\mu}_k = \sum_{j=1}^{N} \mathbb{I}_{(y_j=k)} x_j / \sum_{j=1}^{N} \mathbb{I}_{(y_j=k)}$. However, in high-dimensional spaces, estimating the precision matrix is a challenging task due to the inverse of the empirical covariance matrix being a biased estimator of the precision matrix, and it may be impossible to invert due to numerical issues. To solve this, we utilize shrinkage methods to estimate the precision matrix. To avoid introducing additional computations, we use the empirical Bayes ridge-type estimator (Kubokawa & Srivastava, 2008) to estimate the precision matrix:

$$\widehat{\Sigma^{-1}} = D((N-1)\hat{\Sigma} + tr(\hat{\Sigma})I_D)^{-1}, \tag{4}$$

where $N$ is the number of samples, $D$ is the dimension of the features, and $\hat{\Sigma}$ represents the empirical covariance. Once the parameter estimation is completed, we can input it into Eq. (3) to obtain the weight and bias of the classifier.

Besides the knowledge extracted from visual modality, the prior knowledge of text modality in pre-trained CLIP is calculated by $x_{test}W_c^T$, where $W_c$ is the weights of CLIP's classifier generated from the text encoder by inputting a predefined prompt, such as "a photo of a {class}". For simplicity, we integrate the knowledge from visual and text modalities by mixing the predictions. Therefore, the output logits of the test image are then calculated as:

$$logits = x_{test}W_c^T + \alpha(x_{test}W^T + b), \tag{5}$$

where $x_{test}$ is the visual feature of test image, and $\alpha$ is a hyper-parameter.

### 3.2 EXTENSION TO OTHER SCENARIOS

We further extend our method to base-to-new generalization and unsupervised learning, where our method cannot directly apply, to illustrate the generalization of our method. In order to maintain the simplicity of the method and avoid introducing additional complexities that could impact the assessment of our approach, we only perform straightforward modifications in these two scenarios.

**Extension to Base-to-New Generalization.** For CLIP base-to-new generalization, the model is trained on the base dataset and tested on a new dataset with unseen classes. However, our method cannot be directly implemented in the scenario where data for the new classes is unavailable. Based on the observation that similar samples have similar statistical information (Yang et al., 2021), we propose that our method can be extended to new classes using the KNN algorithm. To achieve

this, we utilize text embeddings of the new classes to query the training set and select the k nearest neighbors as the synthesized labeled data. The process is defined as follows:

$$\tilde{\mathcal{D}}_{new} = \bigcup_{i=K+1}^{M} \{(x,i)|x \in NN^k(t_i, \mathcal{D})\} \tag{6}$$

where $i = K+1, ..., M$ denotes the new classes, $t_i$ denotes its text embedding, and $NN^k(*, \mathcal{D})$ denotes the k-nearest neighbors of training set $\mathcal{D}$. The classifier is then produced utilizing Eq. (3).

**Extension to Unsupervised Learning.** In the unsupervised learning scenario, we only have the unlabeled data $\{x_i\}_{i=1}^N$. Based on the Gaussian assumption in GDA, the unsupervised data $\{x_i\}_{i=1}^N$ follow Gaussian mixture distribution. In order to maintain the simplicity of our method, we directly employ the EM algorithm for estimating the means and covariance matrix. To begin, we initialize the mean vectors and covariance using the zero-shot classifier, assuming equal priors for each Gaussian distribution. In the E-step, we calculate the probability of the unlabeled data $\{x_i\}_{i=1}^N$ as follows:

$$\gamma_{ik} = \frac{\exp(f_k(x))}{\Sigma_{j=1}^{K} \exp(f_j(x_i))}, \tag{7}$$

for the unlabeled data $\{x_i\}_{i=1}^N$, and $f$ is the logit function using Eq. (5). Moving on to the M-step, we update the mean vectors and covariance matrix using the following formulas:

$$\mu_k = \frac{\sum_{i=1}^N \gamma_{ik} x_i}{\sum_{i=1}^N \gamma_{ik}}, \quad \Sigma = \frac{1}{K} \sum_{k=1}^K \frac{\sum_{i=1}^N \gamma_{ik}(x_i - \mu_k)(x_i - \mu_k)^T}{\sum_{i=1}^N \gamma_{ik}}. \tag{8}$$

Subsequently, we update the classifier using Eq. (3) and repeat the EM process until convergence.

# 4 EXPERIMENTS

## 4.1 SETUP

**Dataset.** According to previous works (Radford et al., 2021; Zhou et al., 2022a;b; Zhang et al., 2022), we select 11 publicly available image classification datasets to assess the effectiveness of CLIP few-shot classification, base-to-new generalization, and unsupervised learning. These datasets cover a range of image recognition tasks, including generic object recognition with ImageNet (Deng et al., 2009) and Caltech101 (Li et al., 2004), fine-grained image recognition with OxfordPets (Parkhi et al., 2012), StanfordCars (Krause et al., 2013), Flowers102 (Nilsback & Zisserman, 2008), Food101 (Bossard et al., 2014) and FGVCAircraft (Maji et al., 2013), satellite image classification with EuroSAT (Helber et al., 2019), action classification with UCF101 (Soomro et al., 2012), texture classification with DTD (Cimpoi et al., 2014), and scene recognition with SUN397 (Xiao et al., 2010). Additionally, we also select 4 datasets, ImageNetV2 (Recht et al., 2019), ImageNet-Sketch (Wang et al., 2019), ImageNet-A (Hendrycks et al., 2021b), and ImageNet-R (Hendrycks et al., 2021a), to evaluate the out-of-distribution generalization. Moreover, we adopt 2 commonly used imbalanced datasets, ImageNet-LT (Liu et al., 2019) and Places-LT (Zhou et al., 2017), for CLIP long-tailed classification.

**Training Details.** To align with previous works (Zhou et al., 2022b;a; Zhang et al., 2022), we utilize ResNet-50 (He et al., 2016) as the visual encoder of CLIP for few-shot classification by default. Similarly, following the previous work (Wang et al., 2024), we choose ResNet-101 as the visual encoder of CLIP for imbalanced learning. To evaluate the model's base-to-new generalization and out-of-distribution generalization performance, we followed CoCoOp (Zhou et al., 2022a) and adopted ViT-B/16-based CLIP (Radford et al., 2021). We follow CLIP (Radford et al., 2021) to adopt prompt ensembling on ImageNet and use a single Zero-Shot CLIP on the other 10 datasets. The hyperparameter $\alpha$, which is used to ensemble the classifiers, is searched in the validation set with values ranging from $0.0001$ to $100.0$, and this value is kept constant for new class data. And the $k$ for the KNN algorithm to synthesize the new class dataset is set to 64. All experiments are conducted on a single NVIDIA GeForce RTX 3090. To obtain a reliable estimate of model performance, we conduct three runs with different random seeds and averaged the results.

**Evaluation Protocol.** For the few-shot classification, we adhere to the evaluation protocol proposed by CLIP (Radford et al., 2021). Specifically, we randomly select 1, 2, 4, 8, or 16 instances per

class to form the few-shot datasets. Subsequently, we train our model on the few-shot datasets and evaluate its performance on the full test dataset. For base-to-new generalization, we adopt the standard protocol proposed by CoCoOp (Zhou et al., 2022a). On each dataset, we split the classes equally into two groups, one as base classes and the other as the new classes. The method is trained using only the 16-shot base classes while the evaluation is conducted on base and new classes separately to test generalization ability. Regarding unsupervised learning, we adhere to the evaluation protocol described by UPL (Huang et al., 2022). For imbalanced learning, we split the classes in each benchmark into three groups based on the number of available images per class. These groups are referred to as Many-shot (with more than 100 images), Medium-shot (with 20 to 100 images), and Few-shot (with less than 20 images). We report the accuracy of each group and the macro F1 score.

## 4.2 RESULTS ON FEW-SHOT CLASSIFICATION

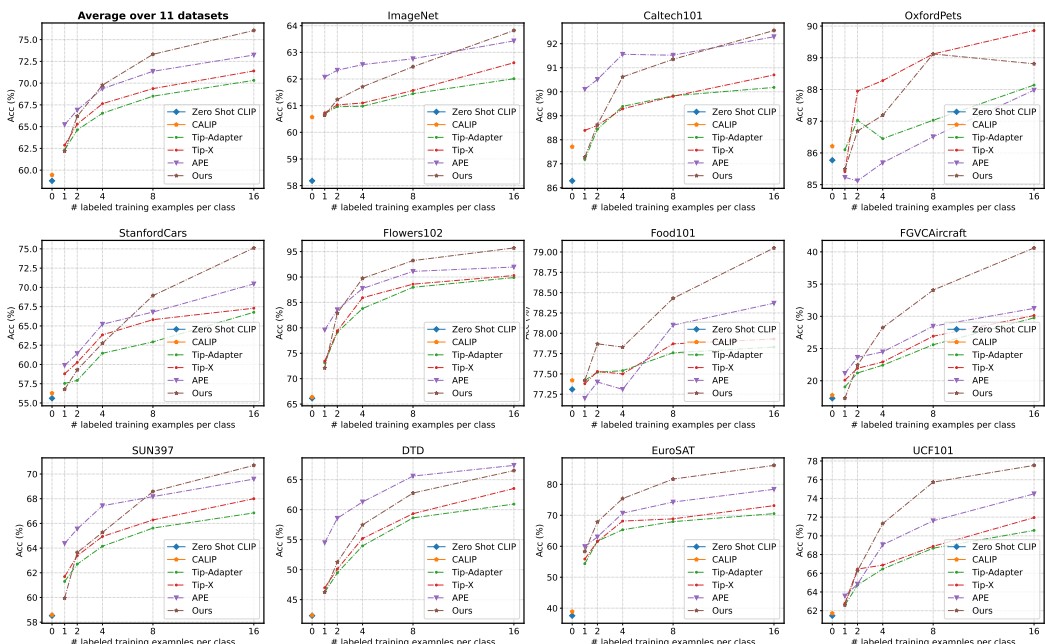

Figure 2: **Results of few-shot classification on the 11 datasets.** We evaluate the performance of our proposed method against five training-free methods under 1, 2, 4, 8, and 16-shot settings. The models are trained using ResNet-50 CLIP. Our method outperforms the baselines significantly.

**Baselines.** We compare our method with two kinds of methods: training-required methods and training-free methods. *For training-required methods*, we consider four baselines: **(1) linear probe** (Radford et al., 2021): Following CLIP, we train a linear classifier on top of high-quality pre-trained CLIP vision encoder's features. **(2) CoOp** (Zhou et al., 2022b): CoOp proposes to learn context prompts for downstream datasets through back-propagation. For comparison, we choose the best version of CoOp with 16 learnable prompts. **(3) CLIP-Adapter** (Gao et al., 2024): CLIP-Adapter proposes to train task-specific adapters to adjust the visual and textual representations. **(4) Tip-Adapter-F** (Zhang et al., 2022): Tip-Adapter-F proposes to build a cache model using the training data to construct the adapter, which is then ensembled with the zero-shot classifier and fine-tuned during training. *For training-free methods*, we consider five baselines: **(1) Zero-Shot CLIP** (Radford et al., 2021): Following CLIP, we build the zero-shot classifier using zero-shot prompts such as "a photo of a {class}". **(2) CALIP** (Guo et al., 2023): CALIP builds a parameter-free attention module to boost CLIP. **(3) Tip-Adapter** (Zhang et al., 2022): Tip-Adaper is the training-free version of Tip-Adapter-F. It builds the adapter with training data and ensembles it with the zero-shot classifier without training. **(4) Tip-X** (Udandarao et al., 2023): Tip-X proposes retrieving images from LAION-5B (Schuhmann et al., 2022) or Stable Diffusion (Rombach et al., 2022) to build the cache of Tip-Adapter. **(5) APE** (Zhu et al., 2023): APE adds a refinement module to Tip-Adapter, which minimizes the inter-class visual similarity and improves the text-image alignment.

**Results.** Figure 2 illustrates the performance of our method and five other training-free baselines: Zero-Shot CLIP (Radford et al., 2021), CALIP (Guo et al., 2023), Tip-Adapter (Zhang et al., 2022),

Table 1: **Results of few-shot classification on 11 datasets.** We report the performance of our method against training-free and training-required baselines on 16-shot datasets. As shown in the table, our method greatly outperforms training-free baselines on average across the 11 datasets and achieves comparable performance as training-required methods. Blue denotes the highest results of training-required methods. **Bold** denotes the highest results of training-free methods.

| Method | Train | Pets | Flowers | FGVC | DTD | EuroSAT | Cars | Food | SUN | Caltech | UCF | ImageNet | Average |
|---|---|---|---|---|---|---|---|---|---|---|---|---|---|
| linear-probe | ✓ | 76.42 | 94.95 | 36.39 | 63.97 | 82.76 | 70.08 | 70.17 | 67.15 | 90.63 | 73.72 | 55.87 | 71.10 |
| CoOp | ✓ | 87.01 | 94.51 | 31.26 | 63.58 | 83.53 | 73.36 | 74.67 | 69.26 | 91.83 | 75.71 | 62.95 | 73.42 |
| CLIP-Adapter | ✓ | 87.84 | 93.90 | 32.10 | 65.96 | 84.43 | 74.01 | 78.25 | 69.55 | 92.49 | 76.76 | 63.59 | 74.44 |
| Tip-Adapter-F | ✓ | 89.70 | 94.80 | 35.55 | 66.55 | 84.54 | 75.74 | 79.43 | 71.47 | 92.86 | 78.03 | 65.51 | 75.83 |
| Zero-Shot CLIP | ✗ | 85.77 | 66.14 | 17.28 | 42.32 | 37.56 | 55.61 | 77.31 | 58.52 | 86.29 | 61.46 | 58.18 | 58.77 |
| CALIP | ✗ | 86.21 | 66.38 | 17.76 | 42.39 | 38.90 | 56.27 | 77.42 | 58.59 | 87.71 | 61.72 | 60.57 | 59.45 |
| Tip-Adapter | ✗ | 88.14 | 89.89 | 29.76 | 60.93 | 70.54 | 66.77 | 77.83 | 66.85 | 90.18 | 70.58 | 62.01 | 70.32 |
| Tip-X | ✗ | 89.86 | 90.29 | 30.12 | 63.53 | 73.12 | 67.30 | 77.93 | 68.00 | 90.70 | 71.95 | 62.61 | 71.40 |
| APE | ✗ | 87.98 | 91.96 | 31.23 | 67.38 | 78.40 | 70.45 | 78.37 | 69.59 | 92.29 | 74.49 | 63.43 | 73.23 |
| Ours | ✗ | 88.81 | 95.72 | 40.61 | 66.51 | 86.12 | 75.12 | 79.05 | 70.70 | 92.55 | 77.53 | 63.82 | 76.05 |

Tip-X (Udandarao et al., 2023), and APE (Zhu et al., 2023), on the 11 downstream datasets, along with their average results. Our method outperforms all of the baselines significantly. Specifically, under the 16-shot setting, our method greatly exceeds Zero-Shot CLIP, CALIP, Tip-Adapter, Tip-X, and APE by 17.28%, 16.60%, 5.73%, 4.65%, and 2.82%. Our method surpasses the baselines on almost all datasets except OxfordPets and DTD.

In Table 1, we further present the numerical results of the training-required and training-free baselines under the 16-shot setting. In the table, our method outperforms training-free methods on 9 of 11 datasets. Specifically, our method achieves a lead of over 3% compared to the second-highest results on Flowers102, FGVCAircraft, EuroSAT, StanfordCars, and UCF101. Moreover, our method achieves comparable performance with training-required methods. And our method achieves great improvement on FGVCAircraft and EuroSAT. This may be because images in these datasets are unusual, where the covariance of the data is important to describe the features.

## 4.3 OUT-OF-DISTRIBUTION GENERALIZATION

We further conduct experiments to assess our method on out-of-distribution generalization. Specifically, we train our model using the 16-shot setting on ImageNet (Deng et al., 2009). Subsequently, we transfer the model directly to target datasets, which included Ima-

Table 2: **Out-of-distribution Generalization.**

| Method | Train | Source | Target | | | | |
|---|---|---|---|---|---|---|---|
| | | IN. | -V2 | -Sk | -A | -R | Avg. |
| CLIP | ✗ | 66.73 | 60.83 | 46.15 | 47.77 | 73.96 | 57.18 |
| CoOp | ✓ | 71.92 | 64.18 | 46.71 | 48.41 | 74.32 | 58.41 |
| Tip-Adapter | ✗ | 70.50 | 63.31 | 48.69 | 50.64 | 77.70 | 60.08 |
| Tip-Adapter-F | ✓ | 73.72 | 65.73 | 48.52 | 49.39 | 77.22 | 60.21 |
| Ours | ✗ | 72.23 | 65.04 | 48.96 | 50.51 | 76.97 | 60.37 |

geNetV2 (Recht et al., 2019), ImageNet-Sketch (Wang et al., 2019), ImageNet-A (Hendrycks et al., 2021b), and ImageNet-R (Hendrycks et al., 2021a).

As presented in Table 2, we choose CLIP, CoOp, Tip-Adapter, and Tip-Adapter-F for comparison. And these methods are based on ViT-B/16-based CLIP. Without requiring any training, our method achieves the highest results on average over the four target datasets. These results indicate that our model is more advantageous in dealing with out-of-distribution generalization and reduces the risk of overfitting on the source dataset.

## 4.4 RESULTS ON IMBALANCED LEARNING

Table 3: **Results of imbalanced learning on ImageNet-LT and Places-LT datasets.** All models are trained on ResNet-101 CLIP. We compare our method with Zero-Shot CLIP (Radford et al., 2021), linear probe, full fine-tune, Balanced Softmax (Ren et al., 2020), CRT (Kang et al., 2020), MARC (Wang et al., 2023a), and their variants with Decoder (Wang et al., 2024).

| Method | Train | ImageNet-LT | | | | | Places-LT | | | | |
|---|---|---|---|---|---|---|---|---|---|---|---|
| | | Many | Medium | Few | Overall | F1 | Many | Medium | Few | Overall | F1 |
| Zero-Shot CLIP | ✓ | 59.57 | 53.57 | 52.81 | 53.62 | 52.50 | 36.23 | 30.43 | **37.89** | 32.17 | 30.85 |
| Linear probe | ✓ | 24.23 | 0.00 | 0.00 | 9.33 | 4.97 | 23.50 | 0.20 | 0.00 | 8.52 | 4.81 |
| Full finetune | ✓ | 74.49 | 52.82 | 26.66 | 57.61 | 55.86 | 47.32 | 28.66 | 11.80 | 32.08 | 30.40 |
| Decoder + Softmax | ✓ | 66.93 | 42.09 | 15.32 | 48.01 | 45.21 | 28.42 | 15.80 | 10.36 | 14.95 | 13.04 |
| Decoder + Balanced Softmax | ✓ | 60.60 | 52.17 | 40.53 | 53.83 | 52.57 | 20.47 | 21.79 | 21.47 | 21.53 | 18.55 |
| Decoder + MARC | ✓ | 58.29 | 54.73 | 46.91 | 55.04 | 54.35 | 12.83 | 25.96 | 27.31 | 25.14 | 22.08 |
| Decoder + CRT | ✓ | 66.89 | 51.98 | 23.82 | 53.89 | 51.77 | 33.14 | 14.70 | 5.01 | 12.77 | 10.78 |
| Full finetune Balanced Softmax | ✓ | 69.18 | 58.25 | 43.63 | 60.47 | 59.79 | 42.42 | 38.41 | 27.93 | 37.81 | 37.21 |
| Full finetune CRT | ✓ | **75.69** | 56.43 | 27.47 | 59.90 | 58.21 | **47.81** | 30.77 | 13.39 | 33.51 | 33.04 |
| Full finetune MARC | ✓ | 73.73 | 58.69 | 32.91 | 60.97 | 59.61 | 46.57 | 38.09 | 17.51 | 37.13 | 35.95 |
| Ours | ✗ | 65.72 | **61.88** | **54.35** | **62.34** | **61.63** | 44.71 | **42.67** | 35.79 | **42.07** | **40.78** |

For imbalanced learning, we compare our method with several imbalanced learning baselines: Zero-Shot CLIP, linear probe, full fine-tuning, and CLIP integrated with specific imbalanced algorithms, namely: MARC (Wang et al., 2023a), CRT (Kang et al., 2020), Balanced Softmax (Ren et al., 2020), and their variants with Decoder (Wang et al., 2024) on ImageNet-LT (Liu et al., 2019) and Places-LT (Zhou et al., 2017) dataset.

The results are shown in Table 3. We report the results in terms of overall accuracy, many-shot accuracy, medium-shot accuracy, and few-shot accuracy, as well as the F1 score. Specifically, we achieve improvements against Zero-Shot CLIP of 8.72% and 9.90% for ImageNet-LT and Places-LT datasets on overall accuracy, respectively. It is worth noting that our method surpasses previous training-required methods, even those trained using imbalanced algorithms. Our method primarily enhances CLIP's performance in terms of medium-shot and few-shot accuracy. This improvement can be attributed to the identical covariance assumption in GDA, which transfers the knowledge of feature distribution from many-shot classes to medium- and few-shot classes.

## 4.5 RESULTS ON BASE-TO-NEW GENERALIZATION

Table 4 presents the results on base-to-new generalization, which show that our approach outperforms the other methods in terms of base accuracy, new accuracy, and their harmonic mean. On average across 11 datasets, our method surpasses CLIP, CoOp, CoCoOp, and KgCoOp by 0.31%, 11.31%, 2.84%, and 0.93% in terms of new accuracy, and by 7.02%, 7.06%, 2.89%, and 1.72% in terms of the harmonic mean. Detailed results on each dataset can be referred to Figure 8 in the Appendix.

Table 4: **Average results over 11 datasets on base-to-new generalization.**

|  | train | base | new | H |
|---|---|---|---|---|
| CLIP | ✗ | 69.34 | 74.22 | 71.70 |
| CoOp | ✓ | 82.69 | 63.22 | 71.66 |
| CoCoOp | ✓ | 80.47 | 71.69 | 75.83 |
| KgCoOp | ✓ | 80.73 | 73.60 | 77.00 |
| Ours | ✗ | **83.96** | **74.53** | **78.72** |

## 4.6 RESULTS ON UNSUPERVISED LEARNING

Table 5: **Results of unsupervised learning.** We compare our method with three baseline methods: Zero-Shot CLIP (Radford et al., 2021), POUF (Tanwisuth et al., 2023) and UPL (Huang et al., 2022).

| Method | Pet | Flo | FGVC | DTD | EuroSAT | Cars | Food | SUN | Cal | UCF | IN | Avg. |
|---|---|---|---|---|---|---|---|---|---|---|---|---|
| CLIP | 85.77 | 66.14 | 17.28 | 42.32 | 37.56 | 55.61 | 77.31 | 58.52 | 86.29 | 61.46 | 58.18 | 58.77 |
| POUF | 88.00 | 66.71 | 16.67 | 41.49 | 42.06 | 57.43 | 74.70 | 58.61 | 86.92 | 61.05 | 55.16 | 58.98 |
| UPL | 88.28 | 68.90 | 17.34 | 46.57 | **54.83** | **62.13** | 77.58 | **63.98** | **89.94** | 67.17 | 60.51 | 63.38 |
| Ours | **89.90** | **72.65** | **18.69** | **46.81** | 49.92 | 60.78 | **78.25** | 63.60 | 87.53 | **68.70** | **61.21** | **63.46** |

In unsupervised learning, the estimation of mean vectors and covariance matrices in GDA is performed by directly applying the EM algorithm for Gaussian Mixture Model (GMM). The results are shown in Table 5. It is noteworthy that this straightforward approach significantly enhances the performance of CLIP in downstream tasks when utilizing unlabeled data. Furthermore, our method consistently outperforms the Zero-Shot CLIP by an average margin of 4.69% across all 11 datasets. Moreover, when compared to the three baseline methods, our approach achieves the highest results on 7 out of the 11 datasets. These results clearly indicate the effectiveness of our method.

| Method | 1 | 2 | 4 | 8 | 16 |
|---|---|---|---|---|---|
| Moore-Penrose | 53.87 | 64.22 | 72.74 | 79.21 | 84.16 |
| EM | 54.28 | 50.85 | 56.86 | 70.69 | 74.35 |
| GraphicalLasso | 57.69 | 64.22 | 71.70 | 76.89 | 76.56 |
| LedoitWolf | **59.59** | 67.01 | 74.23 | 80.52 | 84.33 |
| OAS | 59.58 | 66.81 | 74.25 | 80.46 | 84.31 |
| KS | 58.30 | **67.88** | **75.40** | **81.70** | **86.12** |

Table 6: Comparison of different precision matrix estimation methods on EuroSAT, including Moore-Penrose (Penrose, 1955), EM (Efron & Morris, 1976), GraphicalLasso (Friedman et al., 2008), LedoitWolf (Ledoit & Wolf, 2004), OAS (Chen et al., 2010), and KS (Kubokawa & Srivastava, 2008). The grey color denotes the one used in the paper.

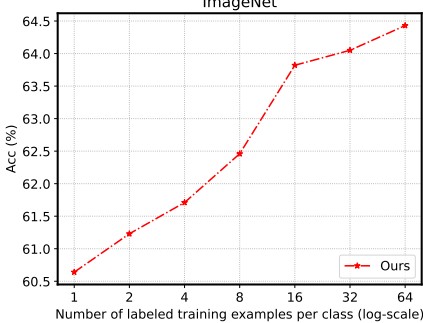

Figure 3: We trained our method on ImageNet with more shots. The x-axis is presented on a logarithmic scale.

### 4.7 ABLATION STUDY

**Effectiveness of Precision Matrix Estimation.** The estimation of the precision matrix is challenging due to the limited data and bias problem. To address this, we employ the empirical Bayes ridge-type estimator (KS) in the paper, which is specifically designed for scenarios where the sample size is smaller than the dimension. We compare it with other robust precision estimation techniques, including Moore-Penrose, the estimator in Efron and Morris (EM), GraphicalLasso, LedoitWolf, and OAS. As shown in Table 6, the empirical Bayes ridge-type estimator achieves the best results, which shows its effectiveness.

**Effectiveness of Increased Sample Size.** We further train our method with more training data. Figure 3 illustrates the results of training our model on ImageNet, using 1, 2, 4, 8, 16, 32, and 64 shots per class. The x-axis is presented on a logarithmic scale. We observe that the model performance increases with an increase in the number of data, and it exhibits a linear relationship with the logarithm of the number of data. This indicates that our approach is not restricted to few-shot learning, but instead has the ability to improve consistently with an increase in the number of samples.

| Method | Acc.(%) | Param. | Train.Set | Train.Time |
|---|---|---|---|---|
| ResNet-50 | 74.2 | 25.6M | full set | > 1 day |
| ResNet-101 | 77.4 | 44.5M | full set | > 1 day |
| DeiT-T | 72.2 | 6.0M | full set | > 1 day |
| DeiT-S | 79.9 | 22.1M | full set | > 1 day |
| Tip-Adapter | 76.1 | 0M | 16-shot | 0 |
| Tip-Adapter* | - | - | full set | - |
| Tip-Adapter-F | 79.4 | 6.2M | 16-shot | 6 min |
| Tip-Adapter-F* | - | - | full set | - |
| Ours | 79.1 | 0M | 16-shot | 1.6 sec |
| Ours | **80.0** | 0M | full set | 3.6 sec |

Table 7: * denotes that the model is out-of-memory. Comparison between Tip-Adapter, Tip-Adapter-F, and conventional methods, ResNet and DeiT, trained by full training set on ImageNet. The training time is tested on a single NVIDIA GeForce RTX 3090.

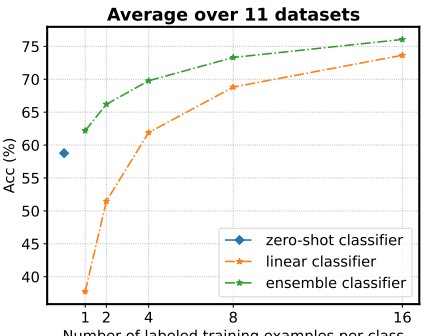

Figure 4: The figure depicts the performance of the CLIP zero-shot classifier, our linear classifier, and their ensemble in the different shot settings on average across 11 datasets.

**Effectiveness of Ensemble Classifier.** We evaluate the effectiveness of the ensemble of linear classifiers, as presented in Eq. (5). Figure 4 shows the performance of the CLIP zero-shot classifier, our linear classifier, and the ensemble classifier in few-shot classification on average on the 11 datasets. We observe that directly using the linear classifier sometimes produces worse results than using the zero-shot classifier. This can be attributed to inaccuracies in the estimated precision matrix, leading to a poor classifier. However, when the classifiers are ensembled according to Eq. (5), the ensemble classifier outperforms both individual classifiers in all settings, demonstrating its effectiveness.

**Comparison to Fully-trained Methods.** In Table 7, we compare efficient fine-tuning methods, Tip-Adapter, Tip-Adapter-F, and our proposed method, with conventional fully trained methods such as ResNet (He et al., 2016) and DeiT (Touvron et al., 2021). We adopt ViT-L/14 CLIP for efficient fine-tuning methods. Although Tip-Adapter and Tip-Adapter-F achieve comparable performance to conventional methods (He et al., 2016; Touvron et al., 2021), they fail to train on full set as they need to cache all the training data, which leads to OOM error. In contrast, our proposed method does not have this problem since we only store the classifier parameters. Therefore, our approach can perform well not only on few-shot but also on the full training set. Furthermore, without requiring any training, our approach achieves the highest performance compared to both efficient fine-tuning methods and conventional training methods.

## 5 CONCLUSION

In this paper, we revisit Gaussian Discriminant Analysis (GDA) with CLIP as a hard-to-beat training-free adaptation method. Without any training, we can directly obtain the classifier from the mean vectors and covariance of the training dataset. We conduct extensive experiments of our method on CLIP few-shot classification and imbalanced learning, and its two simple variants on base-to-new generalization and unsupervised learning. Our method achieves state-of-the-art results against previous training-free methods and is comparable to or even better than training-required methods. These results demonstrate the effectiveness of our method. In the future, we will explore the application of our method in dense prediction tasks and other scenarios such as test-time adaptation.

## REPRODUCIBILITY STATEMENT

In this paper, we provide a comprehensive overview of the datasets, training procedures, and evaluation settings, which are thoroughly discussed in Section 4.1. Detailed statistics for the datasets, prompt templates, and pseudocode can be found in Appendix C. To ensure the reproducibility of our method, we have also made the source code and scripts available in the supplementary materials.

## ACKNOWLEDGEMENT

This work was funded by the National Natural Science Foundation of China under Grant 62276256, the Beijing Nova Program under Grant Z211100002121108, and the Young Elite Scientists Sponsorship Program by CAST.

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

Supplementary Materials Organization:

# A DETAILS OF THE METHOD

## A.1 COMPUTATION OF EQUATION (2)

**Theorem A.1.** *Assuming that the features of different classes follow the Gaussian distribution with identical covariance, i.e., $(X|Y = i) \sim \mathcal{N}(\mu_i, \Sigma)$ for $i = 1, 2, .., K$. Then, the classification probability can be expressed as follows:*

$$p(y = i|x) = \frac{\exp(\mu_i^T \Sigma^{-1} x - \frac{1}{2}\mu_i^T \Sigma^{-1}\mu_i + \log p_i)}{\sum_{j=1}^{K} \exp(\mu_j^T \Sigma^{-1} x - \frac{1}{2}\mu_j^T \Sigma^{-1}\mu_j + \log p_j)}, \tag{9}$$

*Proof.* Since $(X|Y = i) \sim \mathcal{N}(\mu_i, \Sigma)$ for $i = 1, 2, .., K$, the probability of class $i$ is:

$$p(x|y = i) = \frac{1}{(2\pi)^{\frac{d}{2}}|\Sigma|^{\frac{1}{2}}} \exp(-\frac{(x - \mu_i)^T \Sigma^{-1}(x - \mu_i)}{2}), \tag{10}$$

where $d$ is the feature dimension. Later, the classification probability can be derived by using the Bayesian formula,

$$
\begin{aligned}
p(y = i|x) &= \frac{p(x|y = i)p(y = i)}{\sum_{j=1}^{K} p(x|y = j)p(y = j)} \quad \text{(Bayesian formula)} \\
&= \frac{\frac{1}{(2\pi)^{\frac{d}{2}}|\Sigma|^{\frac{1}{2}}} \exp(-\frac{(x-\mu_i)^T \Sigma^{-1}(x-\mu_i)}{2})p(y = i)}{\sum_{j=1}^{K} \frac{1}{(2\pi)^{\frac{d}{2}}|\Sigma|^{\frac{1}{2}}} \exp(-\frac{(x-\mu_j)^T \Sigma^{-1}(x-\mu_j)}{2})p(y = j)} \quad \text{(Using Equation 10)} \\
&= \frac{\cancel{\frac{1}{(2\pi)^{\frac{d}{2}}|\Sigma|^{\frac{1}{2}}}} \exp(\cancel{-\frac{x^T \Sigma^{-1} x}{2}} + \mu_i \Sigma^{-1} x - \frac{\mu_i^T \Sigma^{-1}\mu_i}{2})p(y = i)}{\sum_{j=1}^{K} \cancel{\frac{1}{(2\pi)^{\frac{d}{2}}|\Sigma|^{\frac{1}{2}}}} \exp(\cancel{-\frac{x^T \Sigma^{-1} x}{2}} + \mu_j \Sigma^{-1} x - \frac{\mu_j^T \Sigma^{-1}\mu_j}{2})p(y = j)} \\
&= \frac{\exp(\mu_i^T \Sigma^{-1} x - \frac{1}{2}\mu_i^T \Sigma^{-1}\mu_i + \log p_i)}{\sum_{j=1}^{K} \exp(\mu_j^T \Sigma^{-1} x - \frac{1}{2}\mu_j^T \Sigma^{-1}\mu_j + \log p_j)} \quad \text{(denoted } p_i = p(y = i))
\end{aligned}
\tag{11}
$$

$\square$

## A.2 PSEUDOCODE

**Algorithm 1** Pytorch-like pseudocode for our method.

```
1   # Input:
2   # - X: (N, D) visual features from CLIP visual encoder.
3   # - Y: (N, ) ground-truth label for the features.
4   # - X_test: (M, D) test visual features from CLIP visual encoder.
5   # - Y_test: (M, ) ground-truth label for test features.
6   # - W_c: (K, D) zero-shot classifier generated by prompting.
7   # Output:
8   # - acc: test accuracy.
9
10  def hard_to_beat(X, Y, X_test, Y_test, W_c):
11      # 1. Compute mean vectors for each class.
12      mus = []
13      for i in range(K):
14          idx = torch.where(Y == i)
15          mus.append(X[idx].mean(dim=0))
16      mus = torch.cat(mus)
17
18      # 2. Estimate the precision matrix using Equation (4).
19      # centered features
20      centered_X = torch.cat([(X[torch.where(Y == i)] - mus[i]) for i in range(K)])
21      cov = torch.cov(centered_X)
22      # compute the precision matrix (inverse covariance)
23      inv_cov = D * torch.inv((N - 1) * cov + trace(cov) * eye(D))
24
25      # 3. Compute weight and bias using Equation (3).
26      W = mus @ inv_cov
27      b = log(1 / K) - 0.5 * einsum('nd, dc, nc -> n', mus, inv_cov, mus)
28
29      # 4. Search the hyperparameter using the validation set.
30      alpha = search_hyperparam(W_c, W, b)
31
32      # 5. Test.
33      test_logits = X_test @ W_c.T + alpha * (X_test @ W.T + b)
34      acc = compute_acc(test_logits, Y_test)
35  return acc
```

# B  MORE EXPERIMENTAL ANALYSIS

## B.1  BASE-TO-NEW GENERALIZATION

**Results.** Our method can be extended to the base-to-new generalization scenario by incorporating the KNN algorithm. To accomplish this, we utilize the text embeddings of the new classes to query the training set and select the k nearest neighbors as the training data for the new class. Subsequently, we apply our proposed method to generate the classifier for the new classes using the synthesized dataset. In order to compare our approach, we select CLIP (Radford et al., 2021), CoOp (Zhou et al., 2022b), CoCoOp (Zhou et al., 2022a), and KgCoOp (Yao et al., 2023).

Table 8 presents the results, which demonstrate that our approach outperforms the other methods in terms of base accuracy, new accuracy, and their harmonic mean. On average across 11 datasets, our method surpasses CLIP, CoOp, CoCoOp, And KgCoOp by 14.62%, 1.27%, 3.49%, and 3.23% in terms of base accuracy. It also outperforms them by 0.31%, 11.31%, 2.84%, and 0.93% in terms of

new accuracy, and by 7.02%, 7.06%, 2.89%, and 1.72% in terms of the harmonic mean. Moreover, our approach achieves the highest harmonic mean in 6 out of 11 datasets. These results clearly indicate the effectiveness of our approach in generalizing to new classes.

(a) **Average over 11 datasets**

|  | base | new | H |
|---|---|---|---|
| CLIP | 69.34 | 74.22 | 71.70 |
| CoOp | 82.69 | 63.22 | 71.66 |
| CoCoOp | 80.47 | 71.69 | 75.83 |
| KgCoOp | 80.73 | 73.60 | 77.00 |
| Ours | **83.96** | **74.53** | **78.72** |

(b) ImageNet

|  | base | new | H |
|---|---|---|---|
| CLIP | 72.43 | 68.14 | 70.22 |
| CoOp | **76.47** | 67.88 | 71.92 |
| CoCoOp | 75.98 | **70.43** | **73.10** |
| KgCoOp | 75.83 | 69.96 | 72.78 |
| Ours | 75.95 | 69.79 | 72.74 |

(c) Caltech101

|  | base | new | H |
|---|---|---|---|
| CLIP | 96.84 | 94.00 | 95.40 |
| CoOp | 98.00 | 89.81 | 93.73 |
| CoCoOp | 97.96 | 93.81 | 95.84 |
| KgCoOp | 97.72 | 94.39 | 96.03 |
| Ours | **98.04** | **94.51** | **96.24** |

(d) OxfordPets

|  | base | new | H |
|---|---|---|---|
| CLIP | 91.17 | 97.26 | 94.12 |
| CoOp | 93.67 | 95.29 | 94.47 |
| CoCoOp | **95.20** | 97.69 | **96.43** |
| KgCoOp | 94.65 | **97.76** | 96.18 |
| Ours | 94.10 | 97.15 | 95.60 |

(e) StanfordCars

|  | base | new | H |
|---|---|---|---|
| CLIP | 63.37 | 74.89 | 68.65 |
| CoOp | 78.12 | 60.40 | 68.13 |
| CoCoOp | 70.49 | 73.59 | 72.01 |
| KgCoOp | 71.76 | **75.04** | **73.36** |
| Ours | **78.71** | 66.92 | 72.34 |

(f) Flowers102

|  | base | new | H |
|---|---|---|---|
| CLIP | 72.08 | **77.80** | 74.83 |
| CoOp | 97.60 | 59.67 | 74.06 |
| CoCoOp | 94.87 | 71.75 | 81.71 |
| KgCoOp | 95.00 | 74.73 | **83.65** |
| Ours | **97.78** | 72.46 | 83.24 |

(g) Food101

|  | base | new | H |
|---|---|---|---|
| CLIP | 90.10 | 91.22 | 90.66 |
| CoOp | 88.33 | 82.26 | 85.19 |
| CoCoOp | **90.70** | 91.29 | 90.99 |
| KgCoOp | 90.50 | **91.70** | **91.09** |
| Ours | 90.63 | 91.21 | 90.92 |

(h) FGVCAircraft

|  | base | new | H |
|---|---|---|---|
| CLIP | 27.19 | **36.29** | 31.09 |
| CoOp | 40.44 | 22.30 | 28.75 |
| CoCoOp | 33.41 | 23.71 | 27.74 |
| KgCoOp | 36.21 | 33.55 | 34.83 |
| Ours | **45.88** | 34.09 | **39.12** |

(i) SUN397

|  | base | new | H |
|---|---|---|---|
| CLIP | 69.36 | 75.35 | 72.23 |
| CoOp | 80.60 | 65.89 | 72.51 |
| CoCoOp | 79.74 | **76.86** | 78.27 |
| KgCoOp | 80.29 | 76.53 | 78.36 |
| Ours | **81.95** | 75.62 | **78.65** |

(j) DTD

|  | base | new | H |
|---|---|---|---|
| CLIP | 53.24 | **59.90** | 56.37 |
| CoOp | 79.44 | 41.18 | 54.24 |
| CoCoOp | 77.01 | 56.00 | 64.85 |
| KgCoOp | 77.55 | 54.99 | 64.35 |
| Ours | **80.63** | 59.82 | **68.69** |

(k) EuroSAT

|  | base | new | H |
|---|---|---|---|
| CLIP | 56.48 | 64.05 | 60.03 |
| CoOp | 92.19 | 54.74 | 68.69 |
| CoCoOp | 87.49 | 60.04 | 71.21 |
| KgCoOp | 85.64 | 64.34 | 73.48 |
| Ours | **93.28** | **79.21** | **85.67** |

(l) UCF101

|  | base | new | H |
|---|---|---|---|
| CLIP | 70.53 | 77.50 | 73.85 |
| CoOp | 84.69 | 56.05 | 67.46 |
| CoCoOp | 82.33 | 73.45 | 77.64 |
| KgCoOp | 82.89 | 76.67 | 79.65 |
| Ours | **86.63** | **79.09** | **82.69** |

Table 8: **Base-to-new generalization.** Comparison of CLIP, CoOp, CoCoOp, KgCoOp, and our method. CoOp, CoCoOp, and KgCoOp are training-required methods, while our method is a training-free method. base and new denotes the average accuracy of base and new classes, and H denotes their harmonic mean.

## B.2 ROBUSTNESS TO DIFFERENT ARCHITECTURES

We further evaluate the efficacy of our proposed method across 11 datasets with varying visual architectures of CLIP. We selected two approaches for comparison: a training-required method, CoOp (Zhou et al., 2022b), and a training-free method, Tip-Adapter (Zhang et al., 2022). And these methods are trained on the 16-shot dataset. As shown in Table 9, our method yielded a substantial improvement of 17.28%, 18.20%, 16.18%, and 16.62% on average, compared to the Zero-Shot CLIP (Radford et al., 2021) approach, for ResNet-50, ResNet-101, ViT-B/32, and ViT-B/16 CLIP, respectively, across all 11 datasets. The results demonstrate the effectiveness of our method across different CLIP architectures.

## B.3 ABLATION OF THE HYPER-PARAMETER $\alpha$

As shown in Equation (5), our method needs a hyper-parameter $\alpha$ to integrate the knowledge from visual and text modalities. Specifically, we only performed a coarse search for $\alpha$ within [0.001, 0.01, 0.1, 1, 10, 100]. The search only ascertained the order of magnitude for alpha, providing a foundational understanding of its impact. The optimal alpha values resulting from this exploration

Table 9: **Robustness of different architectures on 11 datasets.** The models are trained under the 16-shot setting with different visual architectures of CLIP. **Bold** denotes the highest results.

| Method | Pets | Flowers | FGVC | DTD | EuroSAT | Cars | Food | SUN | Cal | UCF | IN | Avg. |
|---|---|---|---|---|---|---|---|---|---|---|---|---|
| **ResNet-50** | | | | | | | | | | | | |
| Zero-Shot CLIP | 85.77 | 66.14 | 17.28 | 42.32 | 37.56 | 55.61 | 77.31 | 58.52 | 86.29 | 61.46 | 58.18 | 58.77 |
| CoOp | 87.01 | 94.51 | 31.26 | 63.58 | 83.53 | 73.36 | 74.67 | 69.26 | 91.83 | 75.71 | 62.95 | 73.42 |
| Tip-Adapter | 88.14 | 89.89 | 29.76 | 60.93 | 70.54 | 66.77 | 77.83 | 66.85 | 90.18 | 70.58 | 62.01 | 70.32 |
| Ours | **88.81** | **95.72** | **40.61** | **66.51** | **86.12** | **75.12** | **79.05** | **70.70** | **92.55** | **77.53** | **63.82** | **76.05** |
| **ResNet-101** | | | | | | | | | | | | |
| Zero-Shot CLIP | 86.75 | 64.03 | 18.42 | 38.59 | 32.59 | 66.23 | 80.53 | 58.96 | 89.78 | 60.96 | 61.62 | 59.86 |
| CoOp | 88.57 | 95.19 | 34.76 | 65.47 | 83.54 | 79.74 | 79.08 | 71.19 | 93.42 | 77.95 | **66.60** | 75.96 |
| Tip-Adapter | 87.23 | 90.77 | 31.51 | 62.37 | 66.45 | 72.96 | 81.31 | 67.96 | 93.01 | 73.53 | 64.41 | 71.96 |
| Ours | **91.43** | **96.17** | **42.58** | **68.62** | **86.32** | **79.99** | **82.15** | **72.07** | **93.63** | **79.31** | 66.33 | **78.06** |
| **ViT-B/32** | | | | | | | | | | | | |
| Zero-Shot CLIP | 87.49 | 66.95 | 19.23 | 43.97 | 45.19 | 60.55 | 80.50 | 61.91 | 90.87 | 62.01 | 62.05 | 61.88 |
| CoOp | 88.68 | 94.97 | 33.22 | 65.37 | 83.43 | 76.08 | 78.45 | 72.38 | **94.62** | 78.66 | 66.85 | 75.70 |
| Tip-Adapter | 88.34 | 91.61 | 30.92 | 61.90 | 69.53 | 69.59 | 80.94 | 70.27 | 93.85 | 73.74 | 65.41 | 72.37 |
| Ours | **91.21** | **96.16** | **41.74** | **67.63** | **87.30** | **77.55** | **81.84** | **73.60** | 94.42 | **80.17** | **67.00** | **78.06** |
| **ViT-B/16** | | | | | | | | | | | | |
| Zero-Shot CLIP | 89.21 | 71.34 | 24.72 | 44.39 | 47.60 | 65.32 | 86.06 | 62.50 | 92.94 | 66.75 | 66.73 | 65.23 |
| CoOp | 92.53 | 96.47 | 42.91 | 68.50 | 80.87 | **83.09** | 87.21 | 75.29 | 95.77 | 82.24 | 71.92 | 79.71 |
| Tip-Adapter | 91.54 | 94.41 | 39.48 | 65.68 | 76.58 | 75.44 | 86.47 | 71.85 | 95.10 | 77.94 | 70.46 | 76.81 |
| Ours | **93.73** | **97.92** | **50.33** | **71.26** | **89.19** | 82.63 | **87.27** | **75.87** | **95.79** | **84.09** | **72.24** | **81.85** |

are detailed in Table 10. Notably, our analysis reveals a consistent concentration of optimal alpha values in the range of 1 to 10 across the majority of datasets. Subsequently, we conducted additional sensitivity experiments within this narrowed range. The results of different $\alpha$ values are reported in Table 11. The findings indicate a robust model sensitivity to alpha values, with the exception of $\alpha = 1$. And on average over 11 datasets, the model achieves performance around 75.5.

Table 10: The best $\alpha$ in our method for 16-shot datasets with RN50 CLIP. The best alpha is searched on [0.001, 0.01, 0.1, 1, 10, 100] using the validation set.

| | Pets | Flo | FGVC | DTD | EuroSAT | Cars | Food | SUN | Cal | UCF | IN |
|---|---|---|---|---|---|---|---|---|---|---|---|
| best $\alpha$ | 1.00 | 10.00 | 100.00 | 10.00 | 10.00 | 10.00 | 1.00 | 10.00 | 1.00 | 10.00 | 10.00 |
| Ours | 88.81 | 95.72 | 40.61 | 66.51 | 86.12 | 75.12 | 79.05 | 70.70 | 92.55 | 77.53 | 63.82 |

Table 11: **Ablation on hyper-parameter $\alpha$ of our method.** The models are trained under the 16-shot setting with RN50 CLIP.

| | Pets | Flo | FGVC | DTD | EuroSAT | Cars | Food | SUN | Cal | UCF | IN | Avg |
|---|---|---|---|---|---|---|---|---|---|---|---|---|
| $\alpha = 1$ | 88.81 | 89.32 | 31.10 | 67.65 | 85.87 | 67.54 | 79.05 | 66.73 | 92.55 | 77.53 | 61.79 | 73.45 |
| $\alpha = 3$ | 87.35 | 95.18 | 38.20 | 67.42 | 85.86 | 74.07 | 77.80 | 70.15 | 92.63 | 78.53 | 63.19 | 75.49 |
| $\alpha = 5$ | 86.45 | 95.67 | 39.72 | 67.18 | 85.85 | 75.45 | 76.58 | 70.81 | 90.13 | 78.24 | 63.73 | 75.44 |
| $\alpha = 7$ | 85.71 | 95.90 | 40.36 | 67.00 | 85.82 | 75.82 | 75.76 | 70.97 | 91.97 | 77.84 | 63.91 | 75.55 |
| $\alpha = 9$ | 85.36 | 95.99 | 40.63 | 66.86 | 85.80 | 75.62 | 75.17 | 70.79 | 91.94 | 77.59 | 63.88 | 75.42 |
| $\alpha = 10$ | 85.26 | 95.72 | 40.69 | 66.51 | 86.12 | 75.12 | 74.94 | 70.70 | 91.90 | 77.53 | 63.82 | 75.30 |

## C  EXPERIMENTAL DETAILS

### C.1  STATISTIC OF DATASETS

Following previous work (Zhou et al., 2022a;b; Wang et al., 2023b; Huang et al., 2022; Wang et al., 2024), we conduct experiments on 17 publicly available image classification datasets. The datasets include ImageNet (Deng et al., 2009), Caltech101 (Li et al., 2004), OxfordPets (Parkhi et al., 2012), StanfordCars (Krause et al., 2013), Flowers102 (Nilsback & Zisserman, 2008), Food101 (Bossard et al., 2014), FGVCAircraft (Maji et al., 2013), EuroSAT (Helber et al., 2019), UCF101 (Soomro et al., 2012), DTD (Cimpoi et al., 2014), SUN397 (Xiao et al., 2010), ImageNetV2 (Recht et al., 2019), ImageNet-Sketch (Wang et al., 2019), ImageNet-A (Hendrycks et al., 2021b), ImageNet-R (Hendrycks et al., 2021a), ImageNet-LT (Liu et al., 2019), and Places-LT (Zhou et al., 2017).

Table 12: Detailed statistics of datasets used in experiments.

| Dataset | # Classes | # Training | # Test | Task |
|---|---|---|---|---|
| OxfordPets | 37 | 2,944 | 3,669 | fine-grained pets recognition |
| Flowers102 | 102 | 4,093 | 2,463 | fine-grained flowers recognition |
| FGVCAircraft | 100 | 3,334 | 3,333 | fine-grained aircraft recognition |
| DTD | 47 | 2,820 | 1,692 | Textural recognition |
| EuroSAT | 10 | 13,500 | 8,100 | Satellite image recognition |
| StanfordCars | 196 | 6,509 | 8,041 | Fine-grained car recognition |
| Food101 | 101 | 50,500 | 30,300 | Fine-grained food recognition |
| Sun397 | 397 | 15,880 | 19,850 | Scene recognition |
| Caltech101 | 100 | 4,128 | 2,465 | Object recognition |
| UCF101 | 101 | 7,639 | 3,783 | Action recognition |
| ImageNet | 1,000 | 1.28M | 50,000 | Object recognition |
| ImageNetV2 | 1,000 | - | 10,000 | Robustness of collocation |
| ImageNet-Sketch | 1,000 | - | 50,889 | Robustness of sketch domain |
| ImageNet-A | 200 | - | 7,500 | Robustness of adversarial |
| ImageNet-R | 200 | - | 30,000 | Robustness of rendition styles |
| ImageNet-LT | 1,000 | 115,846 | 50,000 | long-tail object recognition |
| Places-LT | 365 | 62,500 | 7300 | long-tail place recognition |

## C.2 PROMPT TEMPLATES FOR EACH DATASET

For the zero-shot classifier, we employ handcrafted prompts to generate the classifier weight, as proposed in CLIP (Radford et al., 2021). By default, we utilize the prompt template "a photo of {class}." for class labels, where {class} represents the name of the classes. However, for fine-grained classification datasets such as FGVCAircraft (Maji et al., 2013), we incorporate the name of the superclass or a description into the template. The prompt templates for each dataset are shown as follows.

Table 13: Prompt templates for each class.

| Dataset | Prompt template |
|---|---|
| Caltech101 (Li et al., 2004) | "a photo of a {class}." |
| OxfordPets (Parkhi et al., 2012) | "a photo of a {class}, a type of pet." |
| StanfordCars (Krause et al., 2013) | "a photo of a {class}." |
| Flowers102 (Nilsback & Zisserman, 2008) | "a photo of a {class}, a type of flower." |
| Food101 (Bossard et al., 2014) | "a photo of {class}, a type of food." |
| FGVCAircraft (Maji et al., 2013) | "a photo of a {class}, a type of aircraft." |
| SUN397 (Xiao et al., 2010) | "a photo of a {class}." |
| DTD (Cimpoi et al., 2014) | "{class} texture." |
| EuroSAT (Helber et al., 2019) | "a centered satellite photo of {class}." |
| UCF101 (Soomro et al., 2012) | "a photo of a person doing {class}." |
| ImageNet (Deng et al., 2009) | "a bad photo of the {class}." "a origami {class}." "a photo of the large {class}." "a {class} in a video game." "art of the {class}." "a photo of the small {class}." |
| ImageNet-LT (Liu et al., 2019) | "a photo of a {class}." |
| Places-LT (Zhou et al., 2017) | "a photo of a {class}." |

