# OpenReview forum: "A Hard-to-Beat Baseline for Training-free CLIP-based Adaptation"
_ICLR.cc/2024/Conference — ICLR 2024 poster_

### Official Review · Reviewer_e8KM · 2023-10-27

**Soundness:** 3 good
**Presentation:** 3 good
**Contribution:** 3 good
**Rating:** 6
**Confidence:** 5

**Summary:**

This paper proposes a baseline for training-free few-shot adaptation of CLIP models, using a simple Gaussian Discriminant Analysis classifier. When ensembled with the original zero-shot CLIP classifier, the proposed method exhibits good performance, even better than some methods that require training. The proposed method can be extended to base-to-novel/unsupervised scenarios with some specific modifications.

**Strengths:**

- The proposed baseline is very simple and does not require training, which is very important in few-shot transfer scenarios where real-time response is often required.
- If this baseline really has a very good performance, then it will be a good point for rethinking the whole field, which may in turn inspire future research in the right direction.
- The paper is well-written and easy to understand.

**Weaknesses:**

- The most important weakness of this paper is a historical issue that stems from the whole literature: According to a paper also submitted to ICLR 2024 (Benchmarking Few-shot Transferability of Pre-trained Models with Improved Evaluation Protocols, https://openreview.net/pdf?id=O3Mej5jlda), the experiments conducted on the original few-shot transfer benchmark of CLIP are unreliable due to intrinsic design flaws, such as sampling uncertainty, unrealistic hyperparameter selection, etc. The performance can largely fluctuate for more than 10 points, just changing the seed for generating the few-shot task. The authors at least **must** re-evaluate their methods and other compared methods on the **same** dozens (e.g., 50) of randomly sampled few-shot tasks and report the 95% confidence intervals. I will increase my score to acceptance if the re-evaluated results still show that the proposed method obtains better performance with statistical significance.
- The training-free metric-based classifier has been well-studied in the few-shot learning literature. The authors are encouraged to compare GDA with other classifiers, such as the Neareast-Centroid Classifier with Euclidean distance [1] and Mahalanobis distance [2].
- The KNN method for Base-to-Novel generalization requires that the base classes should be very similar to novel classes. The method is expected to perform badly on dataset generalization settings.

[1] Prototypical Networks for Few-shot Learning. NeurIPS 2017.

[2] Improved Few-Shot Visual Classification. CVPR 2020.

**Questions:**

I have no more questions. Please see the weaknesses above.

---

> ### Author Response · Authors · 2023-11-19
> **Response to Reviewer e8KM (Part Ⅰ)**
>
> Dear Reviewer e8KM, we are grateful for your time in reviewing this paper and we provide the point-to-point response to your comments below.
>
> > **[Q1].** The most important weakness of this paper is a historical issue that stems from the whole literature: According to a paper also submitted to ICLR 2024 (Benchmarking Few-shot Transferability of Pre-trained Models with Improved Evaluation Protocols, https://openreview.net/pdf?id=O3Mej5jlda), the experiments conducted on the original few-shot transfer benchmark of CLIP are unreliable due to intrinsic design flaws, such as sampling uncertainty, unrealistic hyperparameter selection, etc. The performance can largely fluctuate for more than 10 points, just changing the seed for generating the few-shot task. The authors at least must re-evaluate their methods and other compared methods on the same dozens (e.g., 50) of randomly sampled few-shot tasks and report the 95% confidence intervals. I will increase my score to acceptance if the re-evaluated results still show that the proposed method obtains better performance with statistical significance.
>
> **[A1].** Thank you for pointing out the potential limitation of the evaluation method in this field. We conducted extensive experiments to assess the validity of the original evaluation method. **In addition, we re-evaluated our method against others on 50 randomly sampled few-shot tasks and reported 95 confidence intervals. The results indicate that our method still obtains better performance with statistical significance.**
>
> After reading the paper [1], we found that it did not provide any experimental evidence indicating a significant impact of randomness on the performance of few-shot learning methods on CLIP. The analysis in the paper[1] (Fig1, Tab1, Tab2) is based on DINOv2 and only on 1-shot EuroSAT dataset. Whether this issue persists in few-shot classification within CLIP remains unanswered.
>
> To address your concerns, we are willing to conduct additional experiments to demonstrate the validity of our method. We first validate whether the original evaluation method is reasonable. Then we re-evaluated our method against others on 50 randomly sampled few-shot tasks.
>
> **(Ⅰ) Sampling Uncertainty.**
> The paper [1] asserts that the CoOp-like evaluation protocol is problematic as *in a single-task scenario, the performance of the method can be largely influenced by the choice of sampled data under few-shot setting*. In the table below, we report the results under different sampled data (i.e., different seeds) in a single-task scenario. All experiments use RN50 CLIP on 16-shot datasets, and we report the corresponding 95 confidence intervals.
>
> From the results, our method and other methods (Tip-Adapter/Tip-Adapter-F) are not severely affected by sampling uncertainty. Moreover, there is no significant difference between random sampling 3 times and random sampling 10 times. This, on the one hand, indicates that Sampling Uncertainty is not a serious problem in few-shot classification with CLIP. On the other hand, it suggests that taking the average of three samples for evaluation has already sufficiently demonstrated the effectiveness of the methods, i.e., the original evaluation protocol is also reasonable.
>
> |Method|Train|Pets|Flo|FGVC|DTD|EuroSAT|Cars|Food|SUN|Cal|UCF|IN|Avg|
> |-|-|-|-|-|-|-|-|-|-|-|-|-|-|
> |CLIP|x|85.77|66.14|17.28|42.32|37.56|55.61|77.31|58.52|86.29|61.46|58.18|58.77|
> |ours (#seed=3)|x|88.81+-0.36|95.72+-0.41|**40.61+-0.38**|66.51+-0.17|**86.12+-0.28**|75.12+-0.41|**79.05+-0.11**|70.70+-0.13|92.55+-0.26|**77.53+-0.64**|63.82+-0.13|76.05+-0.30|
> |ours (#seed=10)|x|**89.46+-0.26**|**95.95+-0.25**|40.55+-0.53|**66.57+-0.60**|85.55+-0.40|**75.30+-0.15**|78.99+-0.06|**70.77+-0.07**|**92.64+-0.20**|77.26+-0.39|63.79+-0.07|**76.07+-0.27**|
> |Tip-Adapter (#seed=10)|x|87.68+-0.31|89.45+-0.28|29.41+-0.47|60.38+-0.43|72.38+-0.99|65.68+-0.27|77.90+-0.04|66.81+-0.12|90.46+-0.23|70.41+-0.56|61.36+-0.06|70.17+-0.34|
> |Tip-Adapter-F (#seed=10)|✓|88.33+-0.40|92.28+-0.58|32.97+-0.47|64.83+-0.65|81.62+-0.79|71.96+-0.57|78.70+-0.60|69.77+-0.06|91.90+-0.24|74.96+-0.87|**64.88+-0.07**|73.84+-0.48|

---

> ### Author Response · Authors · 2023-11-19
> **Response to Reviewer e8KM (Part Ⅱ)**
>
> **(Ⅱ) Performance over sampling 50 few-shot tasks.**
>
> **To further address your concern, we re-evaluated our method against others on 50 randomly sampled few-shot tasks and reported the 95 confidence intervals.** Due to time constraints, we chose CLIP, Tip-Adapter, and Tip-Adapter-F for comparison. All experiments use RN50 CLIP on 16-shot datasets. For each dataset, we sample 50 16-shot tasks with a varying number of classes ranging from 8 to 15. The results are presented in the table below. Without additional training, our method achieved the highest performance in 9 out of 11 datasets and statistically outperformed other methods on average across the 11 datasets.
>
> |Method|Train|Pets|Flo|FGVC|DTD|EuroSAT|Cars|Food|SUN|Cal|UCF|IN|Avg|
> |-|-|-|-|-|-|-|-|-|-|-|-|-|-|
> |CLIP|x|94.88+-0.73|84.94+-2.70|51.43+-2.37|65.71+-1.94|36.55+-0.30|92.78+-1.37|93.78+-0.41|95.47+-0.68|97.23+-0.76|85.38+-1.88|97.35+-0.46|81.41+-1.24|
> |Tip-Adapter|x|95.83+-0.64|97.93+-0.42|63.68+-2.59|80.01+-1.42|72.65+-0.36|95.52+-0.61|94.31+-0.39|96.55+-0.51|97.83+-0.56|91.63+-1.23|97.95+-0.36|89.44+-0.82|
> |Tip-Adapter-F|✓|95.56+-0.55|97.74+-0.48|67.64+-2.33|82.32+-1.37|77.75+-0.54|95.56+-0.72|94.57+-0.49|**97.06+-0.40**|**98.29+-0.37**|92.87+-1.11|97.88+-0.42|90.66+-0.80|
> |Ours|x|**96.36+-0.66**|**99.33+-0.19**|**71.32+-2.35**|**83.96+-1.14**|**86.02+-0.20**|**97.01+-0.46**|**94.77+-0.36**|96.81+-0.52|98.12+-0.47|**94.28+-0.89**|**98.01+-0.34**|**92.36+-0.69**|
>
> **In addition, we have provided the code and complete training logs of our method in the supplementary materials (code/logs/fewshots.log). If you have any further questions about the reliability of our method and experiments, we would be happy to answer them.**
>
> [1] Anonymous, "Benchmarking Few-shot Transferability of Pre-trained Models with Improved Evaluation Protocols", In ICLR 2024 submission.

---

> ### Author Response · Authors · 2023-11-19
> **Response to Reviewer e8KM (Part Ⅲ)**
>
> > **[Q2].** The training-free metric-based classifier has been well-studied in the few-shot learning literature. The authors are encouraged to compare GDA with other classifiers, such as the Neareast-Centroid Classifier with Euclidean distance [1] and Mahalanobis distance [2].
>
> **[A2].** Thank you for the advice. We report the results for Euclidean distance[1] and Mahalanobis distance[2] below. All experiments utilize the RN50 CLIP on 16-shot datasets, and we provide the corresponding 95 confidence intervals. As the results from both evaluation methods are consistent, we continue to use the evaluation method outlined in our paper. For a fair comparison, we also integrate both methods with the zero-shot classifier.
>
> The results are shown below. Despite Mahalanobis distance incorporating covariance, it estimates a covariance matrix for each category, introducing significant noise in few-shot settings. Furthermore, it lacks the use of robust methods for precision matrix estimation, resulting in a precision matrix with substantial noise. Thus, it achieves similar performance with the Euclidean method. As indicated in Table 6 of our paper, the precision matrix estimation significantly influences the final results.
>
> |Method|Pets|Flo|FGVC|DTD|EuroSAT|Cars|Food|SUN|Cal|UCF|IN|Avg|
> |-|-|-|-|-|-|-|-|-|-|-|-|-|
> |CLIP|85.77|66.14|17.28|42.32|37.56|55.61|77.31|58.52|86.29|61.46|58.18|58.77|
> |Euclidean [1]|86.76+-0.89|86.21+-0.34|26.26+-1.22|60.84+-0.54|74.19+-1.42|67.24+-0.04|78.01+-0.07|68.04+-0.54|90.70+-0.34|74.03+-0.32|61.75+-0.05|70.37+-0.52|
> |Mahalanobis [2]|86.77+-0.92|86.32+-0.36|26.43+-1.23|61.19+-0.78|74.44+-1.37|67.45+-0.14|78.01+-0.07|68.21+-0.54|90.80+-0.28|74.16+-0.35|61.77+-0.04|70.50+-0.55|
> |ours|**88.81+-0.36**|**95.72+-0.41**|**40.61+-0.38**|**66.51+-0.17**|**86.12+-0.28**|**75.12+-0.41**|**79.05+-0.11**|**70.70+-0.13**|**92.55+-0.26**|**77.53+-0.64**|**63.82+-0.13**|**76.05+-0.30**|
>
> [1] Jake Snell, et al, "Prototypical Networks for Few-shot Learning." In NeurIPS 2017.
> [2] Bateni, Peyman, et al. "Improved Few-Shot Visual Classification." In CVPR 2020.
>
> > **[Q3].** The KNN method for Base-to-Novel generalization requires that the base classes should be very similar to novel classes. The method is expected to perform badly on dataset generalization settings.
>
> **[A3].** Yes, this is a limitation of the approach. The KNN method is based on the observation that similar classes share similar statistical information [1]. Thus, our method may fail in the case when new classes are fairly dissimilar from base classes. Moreover, as mentioned in our paper, we opted for straightforward modifications to the method to demonstrate its generalizability across diverse settings, rather than developing a complex, yet perfect, solution for each scenario. On the one hand, complex solutions compromise the simplicity of our method; on the other hand, if there are substantial differences in methods across diverse scenarios, it impacts the evaluation of our approach's generalization ability.
>
> [1] Yang, Shuo, et al. "Free lunch for few-shot learning: Distribution calibration." In ICLR 2021
>
>
> We hope these experiments and explanations can address your concerns. It would be appreciated if furthermore response.
>
> With best regards,
>
> Authors of submission 2168

---

> > ### Comment · Reviewer_e8KM · 2023-11-21
> > **Thanks for the detailed rebuttal**
> >
> > Thanks for your comprehensive and detailed rebuttal. All my concerns have been addressed. I thus increase my score to 6.

---

> > > ### Author Response · Authors · 2023-11-21
> > > **Thank you for your endorsement and the increased score.**
> > >
> > > We are delighted to receive your response. We are profoundly grateful for your endorsement of our paper.

---

> ### Author Response · Authors · 2023-11-21
> **Invitation to further discussion**
>
> Dear reviewer,
>
> We genuinely appreciate the time and effort you've invested in reviewing our paper. We have carefully provided relevant responses and results to your concerns. We are eager to further discuss with you and gain your insights **before the end of the Author/Reviewer phase**. Please let us know if any aspect of our work remains unclear or if you have additional feedback.
>
> Thank you.

---

### Official Review · Reviewer_4kGv · 2023-11-01

**Soundness:** 3 good
**Presentation:** 3 good
**Contribution:** 2 fair
**Rating:** 6
**Confidence:** 4

**Summary:**

This paper proposes to revisit a classical algorithm, Gaussian Discriminant Analysis (GDA), and apply it to the downstream classification of CLIP. The method assumes that features belonging to each class follow Gaussian distributions, allowing the classifier to be estimated directly from the data without the need for training. By combining information from both visual and textual modalities, the estimated classifier is ensembled with the original zero-shot classifier within CLIP. The proposed method has been extensively evaluated on 17 datasets, demonstrating its effectiveness in achieving improved classification performance.

**Strengths:**

- The incorporation of Gaussian Discriminant Analysis (GDA) into the CLIP model for training-free downstream tasks is interesting.
- The proposed method is simple yet effective.

**Weaknesses:**

- Though simple yet effective, the overall novelty is limited. The proposed method simply treats the pre-trained CLIP model as a frozen feature extractor, and incorporates the idea of Gaussian Discriminant Analysis (GDA) to estimate the classifier without training for the downstream tasks. The key contribution might simply be the validation of the strong feature extraction capability of the pre-trained CLIP model.
- It lacks of more insightful analyses. While the paper introduces the integration of Gaussian Discriminant Analysis (GDA) into the CLIP features, it falls short in providing a comprehensive and detailed analysis of the underlying mechanisms and the reasoning behind the method's effectiveness.
- The effectiveness of the proposed method is dependent on the hyperparameter α, which controls the trade-off in the classifier ensemble. However, the paper lacks detailed evaluations regarding the sensitivity of this parameter.

**Questions:**

- What is the key novelty of the paper?
- What are the main insights?
- How sensitive is the method w.r.t. the hyperparamter α?

---

> ### Author Response · Authors · 2023-11-19
> **Response to Reviewer 4kGv**
>
> Dear Reviewer 4kGv, we are grateful for your time in reviewing this paper and we provide the point-to-point response to your comments below.
>
> > **[Q1].** What is the key novelty of the paper?
>
> **[A1].** We are sorry we didn't showcase this well in the paper and caused your confusion. Below, we elaborate on the novelty of our paper.
>
> **(Ⅰ) We identify a simple but hard-to-beat training-free classifier.** First of all, all existing efficient fine-tuning methods for CLIP (e.g., CoOp/APE/Tip-Adapter/CLIP-Adapter, etc.) have frozen the feature extractor of the CLIP model. Therefore, we do not simply validate the powerful feature extraction capability of CLIP. **Our first contribution is revealing that there exists a simple but effective classifier that performs on par with or even better than previous complicated methods.** Previous methods either require additional learnable modules, such as learnable prompts, and Adapters, or need extra cache modules to store training data. However, we point out that there exists a simple yet effective training-free classifier (i.e., GDA) that has been overlooked in the existing literature. Without the need for additional optimization and design, it surpasses all training-free methods and achieves results on par with or better than training-required methods. This facilitates rapid model transfer and prompts a reconsideration of the effectiveness of these complex modules, inspiring the future design of more efficient fine-tuning methods.
>
> **(Ⅱ) We extend its applicability to various scenarios.** **Our second contribution is that we further extend our method to other domains with minimal modification, including imbalanced learning, base-to-new generalization, and unsupervised learning.** Our method also achieves superior results in these areas, demonstrating its wide-ranging effectiveness.
>
> > **[Q2].** The insightful analysis of the underlying mechanisms of our method
>
> **[A2].** We attribute the superiority of GDA to its Gaussian assumption. On the one hand, this assumption narrows down the parameter search space, reducing the risk of model overfitting. On the other hand, it introduces covariance into the score calculation (i.e., the $\mu_i^T\Sigma^{-1}x$ term in Eq.5), measuring the anisotropy of space. Previous methods, in contrast, only optimized the similarity between features and corresponding class weights ($w_i^Tx$), overlooking the importance of features in different dimensions. For a comparison, we set the covariance in our method to the identity matrix. The results, as shown in the table below, indicate a significant decline in method performance, yielding results similar to Tip-Adapter.
>
> |Method|Pets|Flo|FGVC|DTD|EuroSAT|Cars|Food|SUN|Cal|UCF|IN|Avg|
> |-|-|-|-|-|-|-|-|-|-|-|-|-|
> |Tip-Adapter|88.14|89.89|29.76|60.93|70.54|66.77|77.83|66.85|90.18|70.58|62.01|70.32|
> |Ours (cov=I)|86.76|86.21|26.26|60.83|74.19|67.24|78.01|68.04|90.70|74.03|61.75|70.37|
> |Ours|**88.81**|**95.72**|**40.61**|**66.51**|**86.12**|**75.12**|**79.05**|**70.70**|**92.55**|**77.53**|**63.82**|**76.05**|
>
> > **[Q3].** How sensitive is the method w.r.t. the hyperparamter $\alpha$?
>
> **[A3].** The following table analyses the sensitivity of the hyperparameter $\alpha$, which has been updated in the supplementary materials of the paper. From the results, it can be observed that our approach is relatively stable with respect to $\alpha$. Except for $\alpha=1$, the average performance across 11 datasets remains around 75.5. Even when all $\alpha$ is set to 10, the model only experiences a marginal drop of 0.7 points, still outperforming CoOp (73.42) and Tip-Adapter (70.32).
>
> |Method|Pets|Flo|FGVC|DTD|EuroSAT|Cars|Food|SUN|Cal|UCF|IN|Avg|
> |-|-|-|-|-|-|-|-|-|-|-|-|-|
> |Ours (alpha=1)|88.81|89.32|31.10|67.65|85.87|67.54|79.05|66.73|92.55|77.53|61.79|73.45|
> |Ours (alpha=3)|87.35|95.18|38.20|67.42|85.86|74.07|77.80|70.15|92.63|78.53|63.19|75.49|
> |Ours (alpha=5)|86.45|95.67|39.72|67.18|85.85|75.45|76.58|70.81|90.13|78.24|63.73|75.44|
> |Ours (alpha=7)|85.71|95.90|40.36|67.00|85.82|75.82|75.76|70.97|91.97|77.84|63.91|75.55|
> |Ours (alpha=9)|85.36|95.99|40.63|66.86|85.80|75.62|75.17|70.79|91.94|77.59|63.88|75.42|
> |Ours (alpha=10)|85.26|95.72|40.69|66.51|86.12|75.12|74.94|70.70|91.90|77.53|63.82|75.30|
> |Ours (paper)|88.81|95.72|40.61|66.51|86.12|75.12|79.05|70.70|92.55|77.53|63.82|76.05|
>
> Hope our response can address your concerns.
>
> With best regards,
>
> Authors of submission 2168

---

> ### Author Response · Authors · 2023-11-21
> **Invitation to further discussion**
>
> Dear reviewer,
>
> We genuinely appreciate the time and effort you've invested in reviewing our paper. We have carefully provided relevant responses and results to your concerns. We are eager to further discuss with you and gain your insights **before the end of the Author/Reviewer phase**. Please let us know if any aspect of our work remains unclear or if you have additional feedback.
>
> Thank you.

---

> ### Author Response · Authors · 2023-11-23
> **End of discussion approaching**
>
> Dear Reviewer,
>
> Since the discussion deadline is approaching in less than 11 hours, we kindly request your feedback on whether the response adequately addresses your concerns. If you have any more questions, we would be happy to provide further clarification.
>
> Your timely response is greatly appreciated.
>
> Thanks.

---

> > ### Comment · Reviewer_4kGv · 2023-11-23
> > **Post rebuttal**
> >
> > The authors have addressed most of my concerns. Introducing covariance into classifier weights is interesting. However, it would be better to discuss more on the computational burden brought by computing covariance. Overall, though the the proposed method is simple, it indeed brings some insights to the community. I would like to raise my score to 6.

---

> > > ### Author Response · Authors · 2023-11-23
> > > **Thank you for your endorsement and the increased score**
> > >
> > > Thank you for increasing the score, and we are pleased to address the remaining concerns.
> > > - The computational burden of calculating covariance is quite small. In Table 7, we present the computation time of our method using the ViT-L/14 CLIP on the 16-shot and full ImageNet datasets. Our approach requires only 1.6 and 3.6 seconds to complete the calculations, which is quite fast.

---

### Official Review · Reviewer_CXou · 2023-11-01

**Soundness:** 2 fair
**Presentation:** 3 good
**Contribution:** 3 good
**Rating:** 6
**Confidence:** 4

**Summary:**

The paper focus on adapting the CLIP model to a target image classification dataset by constructing a classifier on the image features. Here, the classifier is constructed not using gradient based approaches, but weights and bias of the classifier are extracted from the statistics of the dataset. Following Gaussian Discriminant Analysis (GDA), authors assume features of each class to follow Gaussian distribution with identical covariance, thereby express the classifier in terms of class means and covariance extracted from the training dataset. Authors showcase the effectiveness of this method across various different tasks, and results remain competitive to gradient based training methods.

**Strengths:**

-	A simple and effective method.
-	Easy to implement in few lines of code.
-	A novel idea of following Gaussian Discriminant Analysis to improve CLIP performance on downstream tasks.
-	Provided ways to extend the approach to tasks like base-to-new generalization and unsupervised learning.
-	Shown better results among training-free methods and competitive to training based methods.

**Weaknesses:**

Rather than listing weakness, I would like to discuss and clarify few questions. Please see the Questions section.

**Questions:**

-	For completeness, provide reference or formulations on how the numerator of eq. (2) is achieved under the assumption of Gaussian distribution.
-	Do all the datasets contain both validation and test set? How the hyperparameter $\alpha$ is set for datasets without validation set (e.g. ImageNet)?
-	Please report the $\alpha$ value that is used for obtaining the results either in captions or appendix.
-	GDA constructing a linear classifier for CLIP. Discuss why GDA is superior to linear probe? Does linear probe performance depend on the training batchsize? Would increasing batchsize improve linear probe performance?
-	Similarly, how the batchsize effects the results of Tip-Adapter-F?
-	How does the split to base and new classes is made for each dataset?
-	I am skeptical regarding the dataset sampling using KNN for estimating the statistics for new classes. Please discuss the scenarios with new classes that are less similar or dissimilar to the training dataset and how such dataset sampling affect the results.
-	Prior work KgCoop [a] is missing in the base-to-new generalization comparisons.
-	Please report the time required for preparing the classifier (computing mean and precision matrix) in Table 7 under Train time.

[a] Hantao Yao, Rui Zhang, and Changsheng Xu. Visual-language prompt tuning with knowledge-guided 293 context optimization. In Proceedings of the IEEE/CVF Conference on Computer Vision and Pattern 294 Recognition (CVPR), pages 6757–6767, June 2023.

I am willing to revise my rating based on the authors feedback.

---

> ### Author Response · Authors · 2023-11-19
> **Response to Reviewer CXou (Part Ⅰ)**
>
> We thank the reviewer CXou for the positive recommendation and constructive comments. And we provide the point-to-point response to your comments below.
>
> > **[Q1].** For completeness, provide references or formulations on how the numerator of eq. (2) is achieved under the assumption of Gaussian distribution.
>
> **[A1].** Thank you for the advice. We update the calculation process of Equation (2) in the Appendix. Assuming that the features of different classes follow the Gaussian distribution with identical covariance, i.e., $(X|Y=i)\sim \mathcal{N}(\mu_i, \Sigma)$ for $i=1,2,.., K$. The probability of class $i$ is $p(x|y=i) = \frac{1}{(2\pi)^\frac{d}{2}|\Sigma|^{\frac{1}{2}}}\exp(-\frac{(x - \mu_i)^T\Sigma^{-1}(x - \mu_i)}{2})$. Then the classification probability can be expressed as follows:
>
> $$
> \begin{aligned}
> p(y=i|x)
>     & = \frac{p(x|y=i)p(y=i)}{\sum_{j=1}^{K}p(x|y=j)p(y=j)} \quad \text{(Bayesian formula)} \\\\
>     & = \frac{\frac{1}{(2\pi)^\frac{d}{2}|\Sigma|^{\frac{1}{2}}}\exp(-\frac{(x - \mu_i)^T\Sigma^{-1}(x - \mu_i)}{2})p(y=i)}{\sum_{j=1}^{K}\frac{1}{(2\pi)^\frac{d}{2}|\Sigma|^{\frac{1}{2}}}\exp(-\frac{(x - \mu_j)^T\Sigma^{-1}(x - \mu_j)}{2})p(y=j)} \\\\
>     & =
>     \frac{
>     {{\frac{1}{(2\pi)^\frac{d}{2}|\Sigma|^{\frac{1}{2}}}}}
>     \exp
>     ({{{-\frac{x^T\Sigma^{-1}x}{2}}}} + \mu_i\Sigma^{-1}x - \frac{\mu_i^T\Sigma^{-1}\mu_i}{2})p(y=i)}
>     {\sum_{j=1}^{K}{{{\frac{1}{(2\pi)^\frac{d}{2}|\Sigma|^{\frac{1}{2}}}}}}\exp({{{-\frac{x^T\Sigma^{-1}x}{2}}}} + \mu_j\Sigma^{-1}x - \frac{\mu_j^T\Sigma^{-1}\mu_j}{2})p(y=j)}  \\\\
>     & = \frac{\exp(\mu_i^T\Sigma^{-1}x - \frac{1}{2}\mu_i^T\Sigma^{-1}\mu_i + \log p_i)}{\sum_{j=1}^K\exp(\mu_j^T\Sigma^{-1}x - \frac{1}{2}\mu_j^T\Sigma^{-1}\mu_j + \log p_j)} \quad \text{(denoted $p_i = p(y=i)$)}
> \end{aligned}
> $$
>
> > **[Q2].** Do all the datasets contain both validation and test sets? How the hyperparameter $\alpha$ is set for datasets without validation set (e.g. ImageNet)?
>
> **[A2].** All datasets include both validation and test sets except ImageNet. Due to this sole exception and for a fair comparison, we align with previous methods (e.g., Tip-Adapter, APE, etc.) and thus validate $\alpha$ on the test set.
>
> > **[Q3].** Please report the $\alpha$ value that is used for obtaining the results either in captions or appendix.
>
> **[A3].** The $\alpha$ value of 16-shot datasets is attached below, and we have updated it in the appendix.
> ||Pets|Flo|FGVC|DTD|EuroSAT|Cars|Food|SUN|Cal|UCF|IN|
> |-|-|-|-|-|-|-|-|-|-|-|-|
> |$\alpha$|1.00|10.00|100.00|10.00|10.00|10.00|1.00|10.00|1.00|10.00|10.00|
>
> > **[Q4].** Why GDA is better than linear classifier for CLIP.
>
> **[A4].** GDA is superior to linear classifiers mainly because it explicitly makes assumptions about the data distribution. This confers two key advantages. Firstly, it narrows down the parameter search space, mitigating the risk of overfitting to specific local optima. Moreover, optimal parameters within this subspace can be directly obtained through statistics, eliminating the need for gradient optimization. Secondly, under the distribution assumption of GDA, the covariance of the data is explicitly introduced into the score calculation (i.e., the $\mu_i^T\Sigma^{-1}x$ term in Eq.5), while linear classifiers only optimize the similarity between features and corresponding class weights. We posit that the inclusion of covariance, the second-order statistic, is pivotal in this context. We employ the empirical Bayes ridge-type estimator for precision matrix estimation, which is specifically designed for scenarios where the sample size is smaller than the dimension. The results in Table 6 provide an ablation of different covariance matrix estimators.
>
> > **[Q5].** How batch size effects linear probe and Tip-Adapter-F.
>
> **[A5].** The impact of batch size on Linear Probe and Tip-Adapter-F is shown in the table below. All experiments utilize RN50 CLIP on 16-shot datasets. As indicated in the table, smaller batch sizes result in better performance for the Linear Probe. This may be attributed to the fact that in larger batch sizes, the model is more prone to overfitting, while in smaller batch sizes, the model is more likely to escape local optima due to increased randomness. Conversely, for Tip-Adapter-F, a larger batch size yields better results, possibly because it integrates the zero-shot classifier, serving a regularization role. Despite this, there is still a gap between their results and our method.
>
> ||LP(bs=256)|LP(bs=128)|LP(bs=64)|Tip-F(bs=256)|Tip-F(bs=128)|Tip-F(bs=64)|Ours|
> |-|-|-|-|-|-|-|-|
> |Pets|74.87|76.37|77.41|88.90|89.23|88.84|88.81|
> |Flowers|92.02|93.30|94.23|92.34|91.66|90.19|95.72|
> |FGVC|31.17|32.40|33.29|33.37|32.77|31.41|40.61|

---

> ### Author Response · Authors · 2023-11-19
> **Response to Reviewer CXou (Part Ⅱ)**
>
> > **[Q6].** How does the split to base and new classes is made for each dataset?
>
> **[A6].** We adopt the standard evaluation protocol proposed by CoCoOp. For each dataset, we split the classes equally into two groups, one as base classes and the other as the new classes. For example, ImageNet has 1,000 classes. We use the class id in (0-499) as the base classes and (500-999) as the novel classes.
>
> > **[Q7].** Please discuss the scenarios with new classes that are less similar or dissimilar to the training dataset and how such dataset sampling affect the results.
>
> **[A7].** Yes, this is a limitation of the approach. The KNN method is based on the observation that similar classes share similar statistical information [1]. Thus, our method may fail in the case when new classes are fairly dissimilar with base classes. Nevertheless, as mentioned in our paper, we opted for straightforward modifications to the method to demonstrate its generalizability across diverse settings, rather than developing a complex, yet perfect, solution for each scenario. On the one hand, complex solutions compromise the simplicity of our method; on the other hand, if there are substantial differences in methods across diverse scenarios, it impacts the evaluation of our approach's generalization ability.
>
> [1] Yang, Shuo, et al. "Free lunch for few-shot learning: Distribution calibration." In ICLR 2021
>
> > **[Q8].** Prior work KgCoOp [a] is missing in the base-to-new generalization comparisons.
>
> **[A8].** Thank you for pointing out this. We add the comparison into the draft and detailed comparison in Appendix. Average results over 11 datasets are presented below.
>
> ||train|base|new|H|
> |-|-|-|-|-|
> |CLIP|x|69.34|74.22|71.70|
> |CoOp|✓|82.69|63.22|71.66|
> |CoCoOp|✓|80.47|71.69|75.83|
> |KgCoOp|✓|80.73|73.60|77.00|
> |Ours|x|**83.96**|**74.53**|**78.72**|
>
> > **[Q9].** Please report the time required for preparing the classifier (computing mean and precision matrix) in Table 7 under Train time.
>
> **[A9].** Thanks for the advice. We have updated it in the draft. The process is quite fast, it takes about 1.6s on the 16-shot ImageNet and about 3.6s on the full ImageNet dataset.
>
> Hope our response can address your concerns.
>
> With best regards,
>
> Authors of submission 2168

---

> ### Author Response · Authors · 2023-11-21
> **Invitation to further discussion**
>
> Dear reviewer,
>
> We genuinely appreciate the time and effort you've invested in reviewing our paper. We have carefully provided relevant responses and results to your concerns. We are eager to further discuss with you and gain your insights **before the end of the Author/Reviewer phase**. Please let us know if any aspect of our work remains unclear or if you have additional feedback.
>
> Thank you.

---

> ### Author Response · Authors · 2023-11-23
> **End of discussion approaching**
>
> Dear Reviewer,
>
> Since the discussion deadline is approaching in less than 11 hours, we kindly request your feedback on whether the response adequately addresses your concerns. If you have any more questions, we would be happy to provide further clarification.
>
> Your timely response is greatly appreciated.
>
> Thanks.

---

> > ### Comment · Reviewer_CXou · 2023-11-23
> >
> > I appreciate authors for providing detailed response. Most of my concerns are addressed. Meanwhile, I still notice that the proposed method face difficulty with new classes that are less similar or dissimilar to the training dataset. However, I believe that this paper brings new insights and beneficial to the community. Therefore, I keep my score and incline towards acceptance.

---

> > > ### Author Response · Authors · 2023-11-23
> > > **Thank you for your endorsement**
> > >
> > > We're grateful for your appreciation and endorsement. Your review holds significant value for us, and we'll improve it in the future.

---

### Meta-Review · Area_Chair_ceMW · 2023-12-02

**Metareview:**

The authors propose a training-free method to adapt a clip-based vision-language model. The proposed method is based on a classical algorithm, Gaussian Discriminant Analysis (GDA). The proposed method is validated on 17 datasets on few-shot classification, imbalanced learning, and out-of-distribution generalization.
Pros:
* A simple and effective method.
* Better results among training-free methods and comparable to training-based methods.
Cons:
* The proposed method faces difficulty with new classes that are less similar or dissimilar to the training dataset.
* Novelty is a bit limited.

Other more detailed questions raised by all three reviewers are addressed by the authors.
Hence, all three reviewers raised the rating to 6.

**Justification For Why Not Higher Score:**

On the one hand, the proposed method is simple and effective. On the other hand, the novelty is a bit limited.
The results are impressive among training-free methods. However, it is only comparable to training based methods.

**Justification For Why Not Lower Score:**

All three reviewers agree on rating 6 which means the paper is slightly above the acceptance threshold.

---

### Decision · Program_Chairs · 2024-01-16

Accept (poster)